

# Effects of vertical ship exhaust plume distributions on urban pollutant concentration - a sensitivity study with MITRAS v2.0 and EPISODE-CityChem v1.4

Ronny Badeke[1], Volker Matthias[1], Matthias Karl[1], David Grawe[2]

[1]Hereon Institute of Coastal Environmental Chemistry, Helmholtz-Zentrum Hereon GmbH, 21502 Geesthacht, Germany
[2]Center for Earth System Research and Sustainability (CEN), Meteorological Institute, Universität Hamburg, 20146 Hamburg, Germany

*Correspondence to*: Ronny Badeke (Ronny.Badeke@hereon.de)

**Abstract.** The modeling of ship emissions in port areas involves several uncertainties and approximations. In Eulerian grid models, the vertical distribution of emissions plays a decisive role for the ground-level pollutant concentration. In this study, model results of a microscale model, which takes thermal plume rise and turbulence into account, are derived for the parameterization of vertical ship exhaust plume distributions. This is done considering various meteorological and ship-technical conditions. The influence of three different approximated parameterizations (Gaussian distribution, single cell
emission and exponential Gaussian distribution) on the ground-level concentration are then evaluated in a city-scale model. Choosing a Gaussian distribution is particularly suitable for high wind speeds (> 5 m s$^{-1}$) and a stable atmosphere, while at low wind speeds or unstable atmospheric conditions the plume rise can be more closely approximated by an exponential Gaussian distribution. While Gaussian and exponential Gaussian distributions lead to ground-level concentration maxima close to the source, with single cell emission assumptions the maxima ground-level concentration occurs at a distance of about 1500 m
from the source. Particularly high-resolution city-scale studies should therefore consider ship emissions with a suitable Gaussian or exponential Gaussian distribution. From a distance of around 4 km, the selected initial distribution no longer shows significant differences for the pollutant concentration near the ground, therefore model studies with lower resolution can reasonably approximate ship plumes with a single cell emission.

## 1 Introduction

The negative impacts of shipping emissions on human health and the environment remain an ongoing problem in coastal cities. Despite a slowing international maritime trade in 2020 caused by the coronavirus disease, the global commercial shipping fleet grew by 4.1 % in the course of the year, representing the highest growth rate since 2014 (UNCTAD, 2020).

From an air quality perspective, the most problematic combustion products from ship exhaust are oxides of sulfur (SO$_x$), oxides of nitrogen (NO$_x$ = NO + NO$_2$) and particulate matter (PM), followed by carbon monoxide (CO) and volatile organic





compounds (VOC). The exhaust composition depends on the type of fuel, the ship engine and the exhaust gas cleaning
measures (Fridell et al., 2008; Moldanová et al., 2009).

Shipping emissions are usually a local problem and affect the port area, but also heavily populated parts of the city depending
on the urban structure and meteorological conditions. Andersson et al. (2009) found an average contribution of shipping
emissions to the population exposure across Europe of roughly 16.5 % $NO_x$ and 11 % $SO_x$. Huszar et al. (2010) described that

the contribution of ship induced surface $NO_x$ reaches 10–30 % near coastal regions. According to Merico et al. (2017; 2019),
$NO_x$ due to ships and harbor activities could be of a comparable rate to those of road traffic in medium-sized harbor cities, i.e.
up to 40 %. Ledoux et al. (2018) described that harbor emissions contribute to 51 % $SO_2$, 35 % NO and 15 % $NO_2$ of the
average pollutant concentration in the city of Calais, France. In the Hamburg harbor area, Ramacher et al. (2020) modelled an
impact of shipping on the $NO_2$ concentration of around 50 % and between 3 and 30 % in the other parts of the city. They

modelled maximum concentrations of up to 75 µg m$^{-3}$ $NO_2$ close to the port. Bai et al. (2020) described an affected area of 4
–26 km² from ship emissions in Yantian port in the southeastern part of China. Cohan et al. (2011) found an affected area to
be within 2–6 km of the port in San Pedro Bay, California.

Ships at berth are a major contributor to emissions from shipping activities in ports since ocean-going ships consume large
amounts of fuel for heating and electricity (e.g. Hulskotte and Denier van der Gon, 2010). The emissions are also a source of

high uncertainty, as often little is known about the use of auxiliary engines.

Regarding air quality and health, the harmful effects of ship emissions include asthma, lung diseases and cardiovascular
problems. Particulate matter is a significant cause of these diseases (Anderson et al., 2012; Martinelli et al., 2013), but since a
large proportion of it is formed as secondary particles from precursor substances like $SO_2$ and $NO_x$, these gases are also in
focus. Epidemiological and health-related economic studies have been investigating the health effects of ship emissions

intensively over the last 15 years and describing their impact in harbor cities (Andersson et al., 2009; Brandt et al., 2013;
Broome et al., 2016; Corbett et al., 2007; Eyring et al., 2010; Lin et al., 2018; Liu et al., 2016; Ramacher et al., 2019; Sofiev
et al., 2018; Winebrake et al., 2009; Zhang et al., 2019). Shipping emissions also contribute to acidification and eutrophication
of coastal waters by deposition of nitrogen and sulfur compounds (Aksoyoglu et al., 2016; Hunter et al., 2011).

A number of legislative efforts have been made to curb atmospheric pollutant emission from the shipping sector. On January
1 2020, the International Maritime Organization (IMO) enforced the Global Sulfur Cap 2020, according to the revised
International convention for the prevention of pollution from ships (MARPOL) Annex VI which allows a maximum 0.5 %
mass sulfur per mass oil outside of sulfur emission control areas (SECAs). Inside SECAs a maximum of 0.1 % mass sulfur per
mass oil was already enforced from 2015 onwards (MEPC, 2008). The goals can be met, for example, by using cleaner fuels

or exhaust scrubbers. Sulfur dioxide emissions are therefore expected to develop in a beneficial way regarding health and air
pollution levels.

Regarding $NO_x$, the North and Baltic Seas are declared as nitrogen emission control areas (NECAs) since January 1 2021. The
regulation enforces a reduction of $NO_x$ emissions by 80 % compared to the present emission level for newly built ships. This


can be reached by using catalysts (= selective catalytic reduction) or liquefied natural gas (LNG). Karl et al. (2019a) estimated

an 80 % reduction for the entire maritime transport sector to be reached by 2040. The emissions will decrease gradually because nitrogen reduction requirements are only valid for new built ships and an almost full fleet replacement could take more than 30 years. Ramacher et al. (2020) projected a reduction of total premature deaths in the North Sea countries by nearly 1 % by 2030, doubling after 2040. Sofiev et al. (2018) stated that the implementation of the new IMO-2020 policy will cause a global decrease of premature deaths and morbidity due to shipping of 34 % and 54 %, respectively.

Additional measures for reducing emissions include optimizing cruising speed, switching to hydrogen, electricity and wind-assisted propulsion (e.g. Kotrikla et al., 2017; McKinlay et al., 2020; Ramacher et al., 2020).

Ship plumes have been modelled in numerous studies, on various scales and with different approaches. Most existing air quality models run on a regional scale, with a resolution between 4 and 20 km (Hamer et al., 2020). Exemplary for the northern European area, increasing trends in ship emission and reduction measures have been modelled for the North Sea and the Baltic

Sea (Jonson et al., 2015; Matthias et al., 2016; Karl et al., 2019a).

On the urban and microscale a large variety of modeling options for ship emissions exist. Most commonly used for plumes in general are Gaussian dispersion models (e.g. Briggs, 1982; Hanna et al., 1985, 2001) where the pollutant distribution corresponds to a normal probability distribution. Their computational costs are low, however they often assume a steady-state solution, spatially uniform meteorology and straight-line trajectories, making them less suitable for complex air quality

modeling studies. More advanced models used in ship plume studies include Large Eddy Simulations (e.g. Chosson et al., 2008), unsteady Gaussian puff models like CALPUFF (e.g. Poplawski et al., 2011; Jahangiri et al., 2018; Murena et al., 2018; Bai et al., 2020) or Eulerian grid models like EPISODE-CityChem (e.g. Karl et al., 2019b, 2020; Pan et al., 2021; Ramacher et al. 2020) or MITRAS on the microscale (Badeke et al., 2021).

All model studies show a certain degree of over- or underestimations regarding ship emissions. They are partly caused by

assumptions of emission rates when exact engine values are not available as well as inaccurate spatial and temporal emission distribution (Matthias et al., 2018). Overestimations and inaccurate chemical transformation rates can occur if ship emissions are instantaneously diluted in a large grid (von Glasow et al., 2003; Vinken et al., 2011). This error can be reduced by using high-resolution numerical models.

In the Hamburg harbor study from Ramacher et al. (2020), a comparison with measurements revealed an overprediction of

modelled $NO_2$ close to the port area. In their study all shipping emissions were released into the lowest vertical layer of the model (10 m) as area sources on a 1 km · 1 km grid without including effects of plume rise which might have led to the overprediction.

Various studies describe that overestimations and underestimates of modelled emission and concentration values can cancel each other out and assume a general uncertainty of ~30 % in their studies (Broome et al., 2016, Merico et al., 2016, 2017).


The vertical emission distribution has a large effect on modelled concentration values (Pozzer, 2009) as it influences chemical reaction rates and transport processes. For ship emissions, the vertical emission distribution into an Eulerian grid model can





be done by using results of the Ship Traffic Emission Assessment Model (STEAM; Jalkanen et al., 2009, 2012; Johansson et al., 2017). However, it does not include plume rise and has mainly been used for regional studies with large grid cells where

effects of plume rise were neglected and emissions were roughly resolved in the lowest layers (e.g. Karl et al., 2019c; Nunes et al., 2020).

Badeke et al. (2021) pointed out the importance of including plume rise and turbulent downward dispersion when modeling plume concentrations in the near field of a ship. They stated that the effect of wind speed and ship size can cause a pollutant downward dispersion of up to 55 % compared to 31 % in a case without accounting for the obstacle effect. Their results from

the microscale need to be incorporated into a city-scale model to be useable in complex city-scale studies with different emission sources.

Due to the nonlinearity of plume rise and turbulent downwash effects, vertical plume concentration profiles can deviate from the Gaussian shape (Badeke et al., 2021; Bieser et al., 2011; Brunner et al., 2019). However, the sensitivity of an Eulerian model to different initial concentration profile assumptions has not been described.

The aim of this study is to derive advanced vertical concentration profiles for various meteorological and technical conditions for a medium-sized cruise ship. This is done by using the results of a microscale model that include effects of plume rise and downward dispersion. Three parameterizations of different complexity are derived for the vertical concentration profiles and used as emission input profiles in a city-scale model. The sensitivity of ground-level concentration values against different meteorology, surface roughness and selected emission input profiles is evaluated. Finally, recommendations are given towards

which vertical plume parameterization should be used under which meteorological conditions.

## 2 Methodology

The schematic concept of this study is presented in Fig. 1. It is composed of three major parts. First, the obstacle-resolving microscale model MITRAS v2.0 (Grawe et al., 2013; Salim et al., 2018) is used to generate a set of synthetic ship plumes based on technical and meteorological input parameters. This allows studying the impact of obstacle-induced turbulence and

thermal plume rise on the shape of the vertical concentration profile. The shape of these profiles is then parameterized depending on different meteorological and technical input parameters (Fig. 1; Sect. 3). In a second step, the parameterized profiles are used in the city-scale model EPISODE-CityChem v1.4 (Karl et al., 2019b, Karl and Ramacher, 2020) with various meteorological settings and additional terrain information (Sect. 4). Finally, pollution ground-level concentration values at different distances from the source are calculated and the impacts of different plume parameterizations as well as

meteorological input parameters and the surface roughness are compared (Sect. 5).



## 3 Plume parameterization

To better represent microscale effects in the near field of the ship in city-scale models, it is necessary to include the effects of plume rise and turbulence into the vertical emission profile. The first step of this work is to find a good parameterization for

the vertical concentration profile at a point where the plume movement is no longer affected by thermal plume rise or ship-induced turbulence, as these factors are usually not covered in larger-scale models. This is the case at a distance of around 100 m from the source (Badeke et al., 2021). At this distance, the vertical concentration profile is calculated with the microscale model MITRAS (Sect. 3.1). The resulting profiles are parameterized in three different ways, i.e. with a classical Gaussian fit (Sect. 3.2), a very simple single cell emission assumption (Sect. 3.3) and a rather complex exponentially modified Gaussian

fit with an upper plume boundary (Sect. 3.4).

### 3.1 MITRAS

Plume rise and turbulence effects of the source (i.e. the ship) are resolved by running a set of modeling runs with the microscale chemistry, transport and stream model (MITRAS). This non-hydrostatic, three-dimensional Eulerian model is based on the Navier-Stokes equation, the continuity equation and the conservation equations for scalar properties like temperature, humidity

and concentration (Grawe et al., 2013; Salim et al., 2018; Schlünzen et al., 2003, 2018). Since the release of version 2.0 MITRAS has been extended to include radiation calculations for the human thermal environment (Fischereit, 2018), however these extensions are not used in this study.The MITRAS configuration is the same as in Badeke et al. (2021). In short, the highest resolution is 2 m · 2 m · 2 m close to the ship in a domain of roughly 1 km · 1 km horizontally and 500 m vertically. The bottom boundary of the domain is water for which the surface roughness is calculated from the wind speed and for the

presented cases is near zero. No chemical reactions occur in the simulations. A constant high temperature and a vertically directed exhaust velocity is added to the emission cell (i.e. the grid cell above the stack). More detailed information can be found in Badeke et al. (2021).

The input data for ship characteristics and meteorology are presented in Table 1. Default values are constant while for all regression analysis one input parameter at a time is varied along the investigation range. In total, 39 different cases have been

calculated in MITRAS. The corresponding input values are presented in Appendix B1 and B2.

Vertical concentration profiles are derived at a distance of 100 m away from the ship as the mean of a column with 100 m · 100 m cell sizes (see Appendix A1 for the concept). While Badeke et al. (2021) derived a formula for the downward dispersion, i.e. the fraction of concentration that is found below ship stack height, in this study, parameterizations for the whole vertical concentration profiles are calculated that account for near field effects (thermal plume rise and obstacle-induced turbulence).

### 155   3.2 Gaussian scheme

One common way to describe the vertical dilution of a ship plume is to assume a concentration reduction according to a Gaussian curve where the mean value μ corresponds to the height of the central plume axis and the standard deviation σ





describes the vertical strength of diffusion. In this way, high values of σ correspond to a plume with strong vertical diffusion that might be caused by high plume rise mainly due to high exhaust temperatures, low wind speed and/or an instable
atmosphere.

The general formula for a vertical Gaussian profile is:

$$c(h) = \frac{1}{\sqrt{2\pi\sigma^2}} \exp\left(-\frac{(h-\mu)^2}{2\sigma^2}\right), \tag{1}$$

where c is a concentration value and h is the height.

A Gaussian curve was fitted to the results according to a least square minimization with the Levenberg-Marquardt Algorithm
(Moré, 1977). From that, individual values for μ and σ were found.

To parameterize the Gaussian curve, the dependency of μ and σ on meteorological and technical input parameters needs to be investigated. Therefore, single regression analyses have been performed.

To estimate the effect that one single input parameter has on the value of μ or σ, all values but the one of interest for a single regression remain constant at a predefined value. These values were selected according to the previous study of Badeke et al.
(2021), see Table 1.

In the single regressions in Fig. 2, the value of default setting is highlighted with a red asterisk.

As visualized in the regression figures, μ and σ depend on most of the input parameters in an approximately linear way. For the wind speed, a linear correlation for μ and σ has been found against the logarithmic value of $v_{wind}$, which accounts for the natural logarithmic wind profile close to the ground (Prandtl layer). The negative correlation can be interpreted as follows:
Higher wind speed causes the plume to remain at lower altitudes (low plume rise, low μ) and also cause a weaker vertical diffusion (lower σ). The wind speed has the strongest effect on both, μ and σ, within the investigated range of the input parameters.

A different type of linear correlation has been found for the wind angle, which describes the effect of the obstacle (i.e. the ship) orientation towards the wind direction. It ranges from 0° (frontal wind) to 90° (lateral wind). A positive linear dependency
has been found for cos(ϕ) against μ and a negative dependency for cos(ϕ) against σ. This means that frontal wind allows for a higher plume rise (larger value of μ) but a weaker vertical dispersion (lower value of σ) than lateral wind, which can be explained by the stronger turbulent eddies that are created in case of lateral wind (larger obstacle effect). Strong turbulence leads to a strong dispersion but at the same time weakens the plume rise.

Positive linear dependencies have been found for μ and σ against exit velocity and exhaust temperature, which both affect the
initial plume rise.

No clear correlation was found for μ against the atmospheric stability, but a negative dependency has been found for stability against σ. This means that the plume does show stronger vertical dispersion in case of an instable or neutral atmosphere. In a stable atmosphere (i.e. at higher values of Γ), the plume remains narrow as during very stable fanning conditions.





By applying multiple regression analysis (for a more detailed insight into the procedure for ship plume studies see Badeke et
al., 2021), two functions have been determined to parameterize μ and σ based on the meteorological and technical parameters
with all cases in Appendix B1.

$$\mu = 153.54 - 119.48 \log_{10}(v_{wind}) + 4.79 \cos(\phi) + 0.60 \, v_{exit} + 0.075 \, T_{exh} \tag{2}$$

$$\sigma = 57.7 - 41.02 \log_{10}(v_{wind}) - 5.0 \cos(\phi) + 0.41 \, v_{exit} + 0.053 \, T_{exh} - 13.21 \, \Gamma \tag{3}$$

By inserting μ and σ into the Gaussian distribution equation (Eq. 1), individual Gaussian profiles can be determined and used
in larger-scale Eulerian grid models for ship plumes under different meteorological and technical conditions (see Sect. 4.5).

The quality of this parameterization has been tested in two steps. In the first step, the fitting of a Gaussian curve to the original
model with the Levenberg-Marquardt Algorithm (Moré, 1977) has been evaluated. An average fitting quality of $R^2 = 0.92$ has
been found. Especially in cases of strong winds and stable atmospheric conditions, the simple Gaussian distribution delivers
good results. However, in cases of strong plume rise at neutral or instable atmospheric conditions, fitting concentration profiles
with a simple Gauss can result in a poorer fitting quality of $R^2 = 0.8$ (e.g. case # 6 in Appendix B1).

In a second step, the quality of the parameterization was tested against the fitting results, which reached an average of $R^2 = 0.99$. The parameterization can reproduce the fitted curves very well.

For a complete comparison of all investigated cases see Appendix B1 and Table C1.

### 3.3 Single cell emission

A much simpler assumption is that all emission occurs in one emission height. This may be the stack height itself or an effective
emission height, the latter being the case in many simple Gaussian dispersion models that solve plume rise and downward
dispersion analytically.

In Eulerian grid models, the emission height equals the stack height only when the model can account for plume rise due to
hot sources and turbulence due to obstacles, e.g. when using the MITRAS model.

The single cell emission (SCE) assumption used in this model assumes all emission to enter the larger model domain
(EPISODE-CityChem) at the height μ that was calculated by the Gaussian parameterization (Sect. 3.2) from MITRAS results.
In this way, it accounts for plume rise and downward dispersion in a minimalistic way, since the position of the central plume
axis is represented but not the initial dispersion in the first 100−200 m.

### 3.4 Exponentially modified Gaussian scheme with upper plume boundary

The exponentially modified Gaussian distribution (Expgauss) adds an exponential feature to the upper end on the Gaussian
distribution, thereby allowing the curve to be asymmetrical. The concentration function applied here is:

$$c(h) = \frac{\lambda_1}{2} \exp\left(\frac{\lambda_1}{2}(2\lambda_2 + \lambda_1 \lambda_3^2 - 2h)\right) \cdot \mathrm{erfc}\left(\frac{\lambda_2 + \lambda_1 \lambda_3^2 - h}{\sqrt{2}\lambda_3}\right) \tag{4}$$


It contains three shape parameters $\lambda_1$, $\lambda_2$ and $\lambda_3$, as well as the complementary error function erfc(x):

$$\text{erfc}(x) = \frac{2}{\sqrt{\pi}} \int_x^\infty e^{-t^2} \, dt \tag{5}$$

This density function is derived by a convolution of the normal and the exponential probability density functions. Figure 3 gives an impression on how the different shape parameters affect the curve.

$\lambda_1$ is the exponential decay parameter. At $\lambda_1 = 1$ the function resembles an ideal Gauss curve with $\lambda_2 + 1$ as mean and $\lambda_3$ as standard deviation. $\lambda_1 = 0$ results in a constant line. $\lambda_2$ affects the height of the maximum concentration and moves the curve along the y-axis. It resembles the mean value of an ideal Gaussian curve when $\lambda_1 = 1$. $\lambda_3$ determines how steep the non-

exponentially modified part (i.e. heights below the maximum concentration) rises. It also slightly affects the position of the concentration maximum.

As in the case of the Gaussian fit, the Expgauss curve was fitted to the results of the MITRAS simulations according to a least square minimization with the Levenberg-Marquardt Algorithm (Moré, 1977). From that, individual values for $\lambda_1$, $\lambda_2$ and $\lambda_3$ were determined. Next, the meteorological and technical input parameters were plotted against the individual shape parameters

to determine which input affects which shape parameter. Figure 4 shows the corresponding single regressions for the ranges presented in Table 1.

By applying multiple regression analysis based on the results of the strongest single regressions (see Appendix C1 for a comparison of effective ranges), the following parameterizations were found for the shape parameters:

$$\lambda_1 = -0.00445 + 0.002\,v_{wind} - 0.00575\,\Gamma \tag{6}$$

$$\lambda_2 = 77.6 - 52.7\,\log_{10}(v_{wind}) + 2.86\,\cos(\phi) + 0.023\,T_{exh} + 3.86\,\Gamma \tag{7}$$

$$\lambda_3 = 20.4 - 8.28\,\cos(\phi) - 0.0135\,T_{exh} - 6.0\,\Gamma \tag{8}$$

These parameterizations can then be used in equation 4 to calculate the vertical plume profile.

Particularly in cases of a stable atmosphere, the plume rise in the near field tends to be overestimated when fitting with an exponentially modified Gaussian function. MITRAS results show a rather sharp reduction in vertical concentration as soon as

the plume temperature decreases down to ambient temperature. Therefore, an upper plume boundary height $h_{up}$ was calculated based on the MITRAS model results and parameterized similar to the concentration profile functions.

A strong logarithmic dependency of the upper plume boundary on wind speed was found (see Fig. 5, panel a). Larger wind speeds lead to lower maximum elevations that the plume could reach in the near field. A linear dependency was found for the upper plume boundary against exhaust temperature (panel b) and against the function $\text{sgn}(\Gamma)\Gamma^2$ (panel c), which is the square

of the vertical temperature gradient (i.e. stability) where the sign is retained (sign function). See also Badeke et al. (2021, Sect. 3.1.4) for comparable correlations for the stability.

From these regressions, a multiple regression formula was calculated to parameterize the upper plume boundary:

$$H_{up} = 154.09 - 114.0\,\log(v_{wind}) + 0.164\,T_{exh} - 189.0\,\text{sgn}(\Gamma)\Gamma^2 \tag{9}$$



This parameterization was tested against MITRAS model results for a variety of 39 different scenarios (see Appendix B1 and
B2). A correlation of $R^2 = 0.85$ for upper boundary calculation with MITRAS and the parameterization formula was found
(Fig. 5, panel d). The performance is weakest under scenarios of very low plume rise, mainly at high wind speeds (> 10 m s$^{-1}$).
Under these conditions, one can either ignore the upper boundary condition or use the classical Gaussian profile.

Finally, the quality of the Expgauss parameterization from ground to upper plume boundary has been tested in two steps. The
fitting of the Expgauss curve to the original MITRAS results delivered a mean fitting quality of $R^2 = 0.99$ for all 39 investigated
cases, which is better than the Gaussian fit. Furthermore, the quality of the parameterization was tested against the fitting
results, which reached an average of $R^2 = 0.96$. The parameterization can reproduce the fitted curves very well and only shows
weaker results at high wind speeds.

For a complete comparison of all investigated cases see Appendix C1.

Results from the Expgauss parameterization were included into further EPISODE-CityChem calculations from the ground up
to the parameterized upper boundary (Sect. 4.5).

## 4 EPISODE-CityChem

The resulting parameterization for the vertical concentration profile is integrated in the city scale model system EPISODE-
CityChem (Hamer et al., 2020; Karl et al., 2019b). This three-dimensional Eulerian grid model is used to simulate the emission,
transport, dispersion, photochemical transformation and deposition of pollutants on a city-scale. In this study, the focus lies on
the physical distribution of ship plumes and, therefore, gases behave like passive tracers.

### 4.1 Model setup

This section describes the specific setup and inputs selected for this study. A summary is given in Table 2.

The inner part of the city of Hamburg is simulated, representing a northern European harbour city. A horizontal resolution of
100 m x 100 m is used. The overall horizontal domain size is 8 km ·8 km.

The model uses a terrain-following sigma coordinate system defined from an idealized hydrostatic pressure distribution. 30
vertical layers are used with increasing vertical expansion. In the lowest 200 m the vertical resolution is fixed at 10 m. Above
this height, it increases up to a vertical resolution of 250 m. A total height of approximately 1 km is covered. Due to the terrain-
following coordinate system used, this upper limit may vary slightly.

The topography input consists of a 2-dimensional static field of terrain heights that was created using the terrain pre-processor
AERMAP (EPA-454/B-03-003) of the U.S. EPA air dispersion model AERMOD (US-EPA, 2004). It coordinates the
allocation of terrain elevation data from several digitized databases to a user-specified model grid. From the database of
NASA's Shuttle Radar Topography Mission (SRTM, Rodriguez et al., 2005) digital elevation data have been used. They have
a spatial resolution of approximately 100 m and WGS 84 as reference geoid. Digital Elevation Data SRTM3 for the region of
Hamburg have been used with Universal Transverse Mercator (UTM) coordinates of south west corner of the model domain





being x = 559064 and y = 5930727 (UTM zone 32N), which corresponds to 9.89091° E and 53.52215° N, respectively, in Cartesian coordinates. Topography information is converted into landuse classes and surface roughness values distinguishing only two different landuse classes: water and land. Figure 6 shows the investigation area with elevation information and an example plume.

The surface roughness is the height above the displacement plane at which the mean wind becomes zero when extrapolating
the logarithmic wind speed profile downward through the surface layer. For water surfaces, Badeke et al. (2021) used a value close to 0 m depending on the wind speed (Schlünzen et al., 2018). Here, a fixed value of 0.001 m is used, which is reasonable, as the focus lies in the investigation of the behavior of the plume over land. The surface roughness of the land area is varied between 0.1 m and 1.0 m, corresponding to different structures, from low crops to medium-sized building areas (e.g. Wieringa, 1992). It plays a major role in the computation of the friction velocity, the turbulent mixing in the vertical diffusion scheme
and the dry deposition.

The meteorological field is created by the meteorological pre-processor MCWIND v1.2 (Hamer et al., 2020). This software produces a diagnostic wind field based on observational or, as in this study, synthetic data. MCWIND adjusts a first guess wind field to a given topography in such a way that it becomes non-divergent and mass-consistent. The 3D fields are calculated internally by applying surface similarity profiles according to Monin-Obukhov theory.

Horizontal advection is considered using a positive 4$^{th}$ degree Bott scheme (Bott, 1989, 1992, 1993), which calculates flux between grid cells, describing the concentration fluctuations locally. A time-splitting method is employed to solve advection separately in x and y directions.

Vertical advection is solved with an up-stream scheme (Byun et al., 1999), which implicitly assumes that the 3D-wind field is free of divergence. Vertical motion is therefore either convergence or divergence in the input horizontal wind fields. This
allows mass conservation.

Both horizontal and vertical eddy diffusivities are calculated on the Eulerian grid using parameterizations. The horizontal diffusion is calculated using a fully explicit forward Euler scheme (Smith, 1985). The vertical diffusion is solved according to the mixing length theory (Monin-Obukhov similarity theory) by a semi-implicit Crank-Nicholson scheme. Eddy diffusion coefficients are calculated by the urban K(z) method, presented in Hamer et al. (2020). It strongly depends on the surface
roughness, which is one parameter that will be varied in this study.

The transport of pollutants in and out of the model domain is implicitly considered within the 3D advection equations.

The effect of dry deposition is included, whereas the effect of wet deposition is not. The dry deposition is calculated based on the resistance analogy (Simpson et al., 2003).

The EPISODE-CityChem chemistry options on the Eulerian grid include a dispersion without photochemistry (applied here),
a solution for basic NO$_x$-O$_3$ photochemical equilibrium, a detailed two-step urban chemistry solver with 45 gas-phase species (EmChem03-mod) as well as an urban chemistry scheme including heterogeneous gas-phase reactions EmChem09-HET (Simpson et al., 2012; Karl et al., 2019b).





## 4.2 Emission characteristics

Ship emissions are treated by EPISODE-CityChem as an area source. This means that concentrations are diluted instantaneously into the corresponding emission grid cells and emitted pollutants are then subject to advection, diffusion, deposition and chemistry (if activated) in the model grid. The vertical emission distribution corresponds to a parameterized profile derived from MITRAS results (see Appendix A1) for a column of 100 m · 100 m · 10 m downwind from the ship to account for thermal plume rise and obstacle-induced turbulence in the near field. Three different parameterization schemes are

applied.

The first scheme will be standard Gaussian parameterization described in Sect. 3.2. The vertical emission profile was normalized and distributed into the corresponding cells of a vertical column.

In the single cell assumption, the whole emission will be inserted into one single cell of the model. This will be the cell at the height of the mean value in the standard Gaussian parameterization (Sect. 3.3).

The third profile is calculated with the Expgauss parameterization (Sect. 3.4).

A normalized emission rate of 1 g s$^{-1}$ NO$_x$ was selected to easier compare different effects (e.g. different concentration distribution, different meteorology or different surface roughness) on the dispersion.

A NO$_x$ split of 95 % NO and 5 % NO$_2$ is used. However, in this study chemical transformations are not considered and therefore, the ratio will not change, as they will behave as passive tracer gas.

The differences of the chosen parameterization and their effect on the ground-level concentration depending on meteorological conditions and surface roughness will be evaluated. Therefore, the concentration will be calculated with increasing distance from the source along the path of highest ground-level concentration (see Appendix A2 for an exemplary scheme).

## 5 Results and discussion

This section presents pollution ground-level concentration values at different distances from the source. The impacts of

different plume parameterizations as well as meteorological input parameters and the surface roughness on the concentration values are compared and uncertainties are discussed.

## 5.1 Input profile

Three different methods for the initial distribution of vertical plume profiles were presented in Sect. 3. Now the differences of

the resulting ground-level concentrations in dependence of the distance to the source will be examined.

As an example, Fig. 7 shows the initial concentration profiles for the Gaussian and Expgauss profiles based on default input parameters, i.e. a wind speed of 5 m s$^{-1}$ with frontal direction, exit velocity of 10 m s$^{-1}$, exhaust temperature of 300 °C and an atmospheric stability of -0.65 K · 100 m$^{-1}$.





The concentration values in Fig. 7 are normalized, i.e. the vertically integrated emission is 1. In case of single cell emission,

the normalized concentration value is 1 at the height of mean Gaussian distribution. For the exponential Gaussian profile the upper plume boundary of the near field lies at around 200 m. However, this upper boundary is only used for the initial emission distribution. In further EPISODE-CityChem calculations, parts of the plume might rise higher.

These normalized curves are used as initial emission profiles in EPISODE-CityChem according to the vertical resolution of the cells.

As can be seen in Fig. 7, the Gaussian profile tends to distribute a part of the emission to the lowest model layer already in the near field. Therefore, high ground-level concentration values close to the source are to be expected. The exponential Gaussian profile can better represent the plume rise. Therefore, ground-level concentrations will have a maximum at a farther distance. In case of SCE, all emissions occur at the mean height of the Gaussian profile, i.e. with no proportion in the lowest model layer. Therefore, the peak ground-level concentration for the SCE approach occurs several 100 m downwind of the source

position.

Variations of these input profiles can be found in Appendix D1 for several different initial conditions.

### 5.2 Effect of surface roughness

To evaluate the impact of surface roughness on pollutant ground-level concentration, the roughness value for land areas was

varied between 0.1 m (grassland) and 1.0 m (urban area), which was assumed to be the range of surface roughness that can occur in the harbor area. All the remaining input values were kept at default conditions (Table 1). It is important to mention that surface roughness was not included in the calculation of initial vertical plume profiles. Therefore, the initial profiles are all the same as in Fig. 7.

Figure 8 shows the ground-level concentration depending on the distance from the source, the roughness length and the effect

of different initial plume profiles. For all investigated cases, the surface roughness shows larger ground-level concentration values in cases of lower surface roughness. In case of the Gaussian profile, the highest differences occur at a distance of 700 m, where $z_0 = 0.1$ m causes 2.72 µg m$^{-3}$ (113 %) higher ground-level concentration than $z_0 = 1.0$ m. For the SCE assumption, the maximum difference is 1.26 µg m$^{-3}$ (88 %) at 1400 m distance. Finally, for the Expgauss assumption, the highest difference is 2.29 µg m$^{-3}$ (128 %) at 700 m distance.

Decreasing ground-level concentrations in areas of increased roughness lengths (city centers) have also been reported in a model study from Barnes et al. (2014). In their study, the lowest model layer experienced localized high ground-level concentration values of $NO_x$ in a city center where the main source of $NO_x$ is traffic. They expected a low ground-level concentration at high surface roughness due to weaker horizontal ventilation but the turbulent mixing effect dominated, thus causing lower ground-level concentrations when modeling with higher surface roughness. In our study, increased dilution also

causes lower concentration values when the surface roughness is high.





When comparing the effect of initial plume profile, it can be seen that the highest ground-level concentrations occur close to the source when assuming a Gaussian distribution (see also Appendix D2). The SCE assumption shows a rather flat maximum between 1000 m and 2000 m, while the Expgauss distribution shows a similar behavior as Gaussian distribution but with a

smaller maximum close to the source. This can all be attributed to the ratio of emission that is initially distributed into the lower modeling layers (see Fig. 7). In case of SCE, all emissions enter the modeling domain at a height of around 100 m and need a much longer distance to be transported downward. At a distance of around 1500 m, the ground-level concentration becomes independent of the initial plume profile.

### 5.3 Effects of stability


Stability effects on the ground-level concentration were tested in EPISODE-CityChem for three different temperature profiles: the standard atmosphere stability ($-0.65 \text{ K} \cdot 100 \text{ m}^{-1}$), a very stable atmosphere ($0.0 \text{ K} \cdot 100 \text{ m}^{-1}$) and an instable atmosphere ($-1.2 \text{ K} \cdot 100 \text{ m}^{-1}$). Again, all the remaining input values were kept at default conditions (Table 1).

Figure 9 displays the effect of stability on the ground-level concentration for the different input concentration assumptions.

For all cases, the highest ground-level concentrations are reached in case of instable atmospheres. This is especially strong in the nearest 1000 m. Here, the obstacle effect of the ship causes stronger turbulent mixing at high instability and more downward dispersion. Under stable conditions the downward transport is weak and the largest proportion of the concentration remains at emission height.

For Gaussian and Expgauss profiles the strongest absolute difference of $3.16 \text{ µg m}^{-3}$ (241 %) and $2.00 \text{ µg m}^{-3}$ (378 %),

respectively, occurs at a distance of 200 m from the source when comparing the instable and the very stable cases. For the SCE assumption, the highest absolute difference of $1.45 \text{ µg m}^{-3}$ (302 %) occurs at a farther distance of 900 m when comparing instable and very stable case.

At a distance of more than 3 km, the difference in ground-level concentration between different stabilities and input profiles are almost negligible. At this distance and slow mixing due to wind and surface roughness effects cause rather even results.

Note that this is only valid for the default settings.

### 5.4 Effects of wind speed

EPISODE-CityChem simulations with six different wind speeds have been performed: $1 \text{ m s}^{-1}$, $2 \text{ m s}^{-1}$, $3 \text{ m s}^{-1}$, $5 \text{ m s}^{-1}$, $8 \text{ m s}^{-1}$ and $12 \text{ m s}^{-1}$. This covers a typical range of values in Northern European harbor cities (see Appendix A in Badeke et al., 2021).

All the remaining input values were kept at default conditions (Table 1).

Results of ground-level concentration simulations are presented in Fig. 10. The effect of different wind speeds on the ground-level concentration is a complex phenomenon in this study. Highest concentration values are found at $1 \text{ m s}^{-1}$, then there are minimum values between $3 \text{ m s}^{-1}$ and $5 \text{ m s}^{-1}$ while at higher wind speeds, the ground-level concentration rises again.





Two different effects cause this behavior. For slow wind speeds, advective transport is low and the pollutants accumulate to a
higher rate. This alone would lead to the impression that ground-level concentration are lowest at high wind speeds. However,
a second effect increases the ground-level concentration with increasing wind speed. This is caused by the input concentration
profile, which shows a weaker plume rise and a stronger obstacle-induced downward dispersion at high wind speeds (see
Appendix D1 for different input profiles depending on wind speed). Badeke et al. (2021) described high wind speed as the
most important factor for downward dispersion of the plume. This is caused by a) strong turbulent eddies formed in the wake
of the ship which dilute it and b) a weaker thermal plume rise at higher wind speeds, as the plume is transported faster and
thus cools down more quickly.

Ledoux et al. (2018) also found higher concentrations with increased wind speed and described that low wind speeds rather
lead to a vertical dispersion and lower concentrations.

Comparing the different parameterizations, the Gaussian profile shows the strongest differences of 9.12 µg m$^{-3}$ (374 %) when
comparing 1 m s$^{-1}$ and 3 m s$^{-1}$ wind speed at a distance of 200 m. The SCE assumption shows a similar maximum absolute
difference of 9.63 µg m$^{-3}$ but with a much larger corresponding relative difference (1095 %) when comparing 1 m s$^{-1}$ and 5 m
s$^{-1}$ at a distance of 600 m. Finally, the Expgauss profile shows a maximum absolute difference of 9.73 µg m$^{-3}$ (506 %) at 600
m when comparing 1 m s$^{-1}$ and 5 m s$^{-1}$.

At a distance of more than 1500 m, the individual plume profiles show very similar results.


## 5.5 Comparison of the effects

Table D1 summarizes the results of Sect. 5.2 to 5.4 and allows a comparison of the effect of different input variables on the
ground-level concentration. Under default conditions, the strongest effect was found for wind speed variations, causing
differences > 9 µg m$^{-3}$ or up to over 1000 %. Stability and roughness length can both cause differences in the range of 1 to 3
µg m$^{-3}$ under default conditions. The strongest differences between the different input parameterizations occur in the first 1500
m from the source. Gaussian profiles give the best representation at high wind speeds and when downward dispersion near the
source are strong. Expgauss profiles can better account for instable atmospheres and strong plume rise. The SCE approach is
simple but always leads to a ground-level concentration maximum at a larger distance from the source (i.e. around 1000 m
downwind). This approach is certainly not optimal when measurements close to the source are underestimated.


## 5.6 Discussion of uncertainties

The performance of MITRAS has been verified before with quality ensured wind tunnel data, including simple obstacle
configurations and results showed a very good agreement of the wind field for most test cases (Grawe et al., 2013). The plume
rise effects have been compared to the integral plume rise model IBS-PLURIS (Janicke and Janicke, 2001) in Badeke et al.
(2021). The initial plume rise was generally some meters higher in the MITRAS study, as MITRAS accounts for the change



in the thermodynamic field and the heat balance equation creates additional buoyancy that is not accounted for in simple Gaussian approaches.

The performance of EPISODE-CityChem has been evaluated in Karl et al. (2019b) with a series of statistical tests, including comparisons against the standard EPISODE model, the air pollution model (TAPM, Hurley et al., 2005; Hurley, 2008) and

measurements in the city of Hamburg. It fulfills the model performance objectives set for the Air Quality Directive, which qualifies it for use in policy applications. From these previous performance evaluations it is assumed that the model setup in this study is capable of reproducing ship plume scenarios in a realistic manner.

In the EPISODE-CityChem part of this study, the ship plume is classified as an area source and not a point source as in the majority of plume model studies (e.g. Poplawski et al., 2011; Merico et al., 2019; Bai et al., 2020; Pan et al., 2021). This can

lead to a poorer performance at lower grid resolutions when the emission is instantaneous diluted equally into the corresponding emission cells (e.g. Huszar et al., 2010, Vinken et al., 2011, Jonson et al. 2015). Studies that treat ship emissions as area sources are rather rare (Kotrikla et al., 2013; Abrutytė et al., 2014). However, in this study the emission profile is adjusted based on MITRAS parameterizations of the initial plume distribution, accounting for plume rise and obstacle-induced turbulence in the near field. The MITRAS results are based on a point source approach, where emissions enter the grid in a 2

m · 2 m · 2 m grid (Badeke et al., 2021). Afterwards the plume concentration profile is used for the vertical emission distribution in the EPISODE-CityChem model. At a distance of roughly 100 m from the source, a dilution of the plume to a 100 m · 100 m area source can be considered acceptable and the equally dilution error is further reduced when applying Gaussian or Expgauss vertical distribution.

Applying Monin-Obukhov similarity theory for the vertical wind profile in the surface layer has a limitation for models with

a high vertical resolution. The logarithmic wind profile is inaccurate inside the surface roughness layer in cases where the surface roughness is not considerably smaller than the lowest model layer height and the wind speed then tends to be overestimated (e.g. Lee et al., 2020). Basu and Lacser (2017) presented an overview on this issue recommending the modeling community to follows a guideline of z1 > 50 $z_0$, where $z_1$ corresponds to the lowest model layer height. This condition is not fully satisfied in some of the EPISODE-CityChem simulations with higher surface roughness. However, EPISODE-CityChem

includes empirical stability correction functions for the surface layer wind profile that address this problem (Holtslag, 1984; Holtslag and de Bruin, 1988) and from the results herein, no evidence of inaccuracies in the plume dispersion even at higher surface roughness was found.

Input assumptions are based on a medium-sized cruise ship with a stack height of approximately 50 m. The selected range of input values such as exit velocity and exhaust temperature have already been discussed by Badeke et al. (2021). For smaller

ships the distribution curves can vary. An adjustment to different stack heights is possible and in a first approximation done by shifting the emission distribution by the difference of the chimney heights.

A complete validation of the vertical profiles is only possible by comparing them with real measurements that also need the inclusion of correct emission factors, other sources and chemistry effects. A precise estimation of emission factors of moored ships includes a further uncertainty, namely the inaccurate data basis for the use of auxiliary engines that are used during




hoteling. Most studies investigate emissions of main engines, while only a few specifically measured or modelled auxiliary
engines (e.g. Abrutytė et al., 2014; Cooper 2003; Eyring et al., 2005; Moreno-Gutiérrez et al., 2015; Tzannatos, 2010). Large
ships have generally between three and seven auxiliary engines (Jayaram et al., 2011) and uncertainties arise from individual
engine operating days, engine load, the specific fuel consumption and the kind of performed operation, e.g. hoteling or loading
(Cooper 2003; Moreno-Gutiérrez et al., 2015). Including all of this was beyond the scope of this study but the inclusion of the
new emission profiles into a more complex chemistry transport model study is planned for the future.

**6 Conclusion**

This study served to improve the modeling of ship exhaust gases on the city scale with regard to the vertical pollutant
distribution. In a first step, vertical concentration profiles were calculated using the microscale MITRAS model, which takes
into account plume rise and obstacle turbulence. This was done for various meteorological and ship-technical conditions to
cover a variety of possible scenarios in Northern Europe harbor cities. From the MITRAS results, three different
parameterizations for the emission distribution in the city-scale model EPISODE-CityChem were derived. Their effect on the
urban ground-level concentration have been compared under conditions of varying urban surface roughness, wind speed and
atmospheric stability.
Based on the model results of this study, the authors would like to make recommendations on which vertical plume
parameterization should be used and when. A general differentiation is recommended for studies with horizontal resolutions ≤
4 km, i.e. especially on the city-scale. At a larger distance from the source, the profiles deliver approximately the same results
for pollution close to the ground. Therefore, a simple single cell emission is sufficient for open ocean or regional studies with
horizontal resolutions > 4 km. Note that the emissions should still be inserted into the correct vertical cell. Equation (2) is then
used to calculate the emission height.
At smaller scales, authors recommend the use of a Gaussian profile in case of moderate or strong wind speeds (> 5 m s⁻¹) and
neutral to stable atmosphere (Γ > -1.0 K · 100 m⁻¹). Regression results for the parameterization were close to R² = 1.0 in these
cases. A vertical Gaussian distribution for stable boundary layers has also been applied for a ship emission study with
AERMOD (Cohan et al., 2011). Gaussian parameterization can also be recommended in case of moving ship studies, e.g. Pan
et al. (2021) since ship speed and wind speed often sum up to a higher effective wind speed, which should then be used in the
parameterization formulae.
For calm wind situations or instable atmospheres, which can occur in harbors under hoteling situations, the Expgauss
parameterization can better account for the initial plume rise and is recommended. A comparable result is expected by applying
the Bi-Gaussian distribution, which is used in case of convective boundary layers in AERMOD (e.g. Cohan et al., 2011).
To the authors' knowledge, this approach is the first of its kind to develop a dynamic vertical emission profile for ship
emissions, including effects of plume rise and downward dispersion and it allows an adjustment of the emission profile for



each time step. This is especially useful in cases of moving sources where the ship orientation and flow angle change frequently. A future study is planned to combine results of this study with the moving point source approach from Pan et al. (2021) for the EPISODE-CityChem modeling system. This will allow a time-flexible variation of the vertical profile of
shipping emissions with either the Gaussian or Expgauss profiles derived here.

**Author contribution**

Conceptualization: all authors; methodology and plume parameterization: R. Badeke and D. Grawe; EPISODE-CityChem: R. Badeke and M. Karl; discussion and conclusion: all authors; writing: R. Badeke. All authors have read and agreed to the
published version of the manuscript.

**Code and data availability**

Currently the MITRAS source code is distributed upon request under the terms of a user agreement with the Mesoscale and Microscale Modeling (MeMi) working group at the Meteorological Institute, University of Hamburg (https://www.mi.uni-hamburg.de/memi). A copy of the user agreement is available upon request. Due to current copyright restrictions, users are
requested to contact the corresponding authors to obtain access to the code free of charge for research purposes under a collaboration agreement (metras@uni-hamburg.de). Documentation for the M-SYS model system in which MITRAS is included, is available online at https://www.mi.uni-hamburg.de/memi under "Numerical Models".

The source codes of the EPISODE-CityChem model version 1.4 and the pre-processing utilities are accessible in release under the RPL license at https://zenodo.org/record/3862264 (Karl and Ramacher, 2020). Pre-processing tools for EPISODE-
CityChem are written in Fortran 90. Software requirements for the utilities and the EPISODE-CityChem model are installation of the gcc/gfortran fortran90 compiler (version 4.4. or later) and the netCDF library (version 3.6.0 or later).

Regression data and a Python scripts for calculating the different vertical plume profiles as well as $h_{up}$ have been added as supplementary material and uploaded at https://doi.org/10.5281/zenodo.5675747 (Badeke, 2021).

**Conflict of interest**

The authors declare no conflict of interest. The funders had no role in the design of the study; in the collection, analyses, or interpretation of data; in the writing of the manuscript, or in the decision to publish the results.



**Acknowledgment**

This work was funded by the German Science Foundation (DFG) in the framework of DFG-NSFC funded project ShipChem.
The authors would kindly like to thank Prof. Dr. Bernd Leitl and Prof. Dr. Kay-Christian Emeis for the fruitful discussions
during the preparation of this manuscript.

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





## Appendix

### A: Plume evaluation schemes

Figure A1 describes the scheme after which vertical concentration profiles from MITRAS have been derived. These concentration profiles were later normalized and used as vertical emission profiles in EPISODE-CityChem.

Figure A2 describes the scheme for deriving ground-level concentration versus distance plots.

### B: Gauss and Expgauss statistics

Tables B1 and B2 present the results of the Gaussian and Expgauss regression analyses, based on which Eq. (2), (3) and (6)−(9) have been derived.

### C: Single regression range-table

Table C1 presents a quantitative comparison of how strong the different input parameters affect the shape parameters for 775 Gaussian and Expgauss fits.

### D: Comparison of parameterizations

Figure D1 presents different input profiles for the EPISODE-CityChem simulation part of this study and the effect of atmospheric stability and wind speed on the profile shape. Figure D2 compares ground-level concentration values depending on the distance to the source for different settings and initial profiles. Table D1 summarizes the results of Sect. 5.2 to 5.4 and 780 allows a comparison of the effect of different input variables on the ground-level concentration. Maximum absolute concentration differences ($\Delta c_{max}$) for the individual variable (surface roughness, stability, wind speed) and initial emission profile (Gauss, SCE, Expgauss) are presented.





**Figures and Tables**

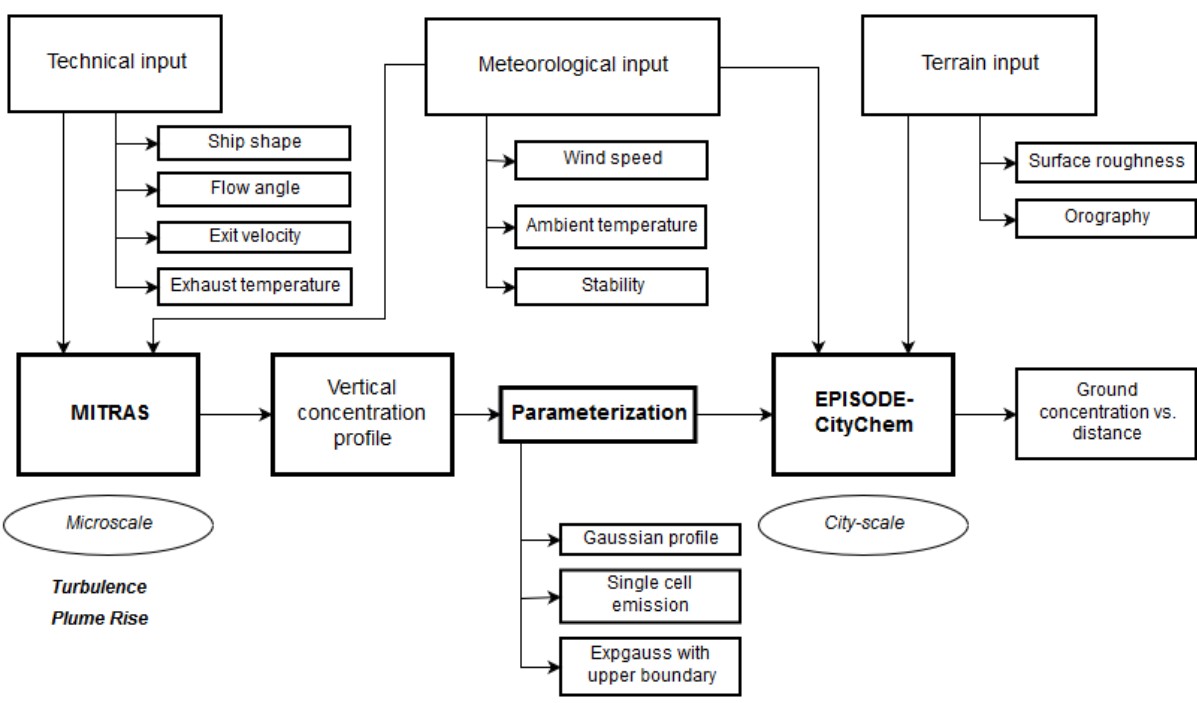

**Figure 1: Schematic concept of this study.**



**Figure 2: Single regression analysis of µ and σ against the input variables wind speed ($v_{wind}$), flow angle ($\phi$), exit velocity ($v_{exit}$), exhaust temperature ($T_{exh}$) and atmospheric stability ($\Gamma$).**


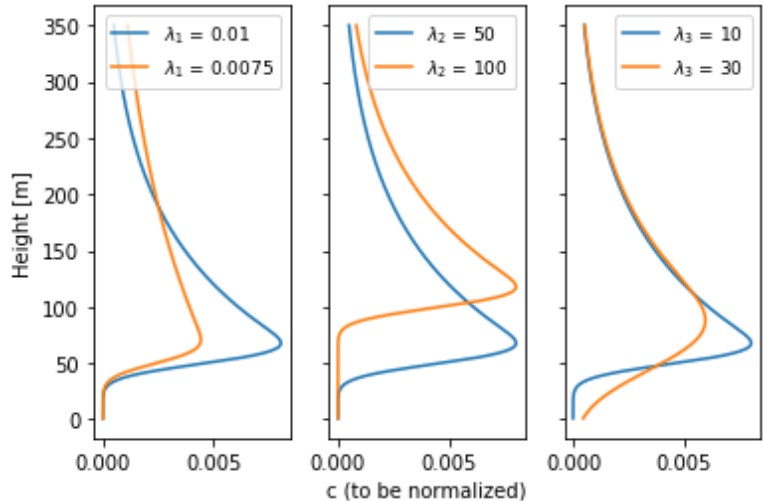

**Figure 3: Visualization of the effect of different shape parameters on the exponentially modified Gaussian distribution. Blue profiles are the same with $\lambda_1 = 0.01$, $\lambda_2 = 50$ and $\lambda_3 = 10$, while in the orange profiles one shape parameter is varied in each panel.**






**Figure 4: Single regression analysis of λ₁, λ₂ and λ₃ against the input variables wind speed ($v_{wind}$), flow angle (φ), exit velocity ($v_{exit}$), exhaust temperature ($T_{exh}$) and atmospheric stability (Γ).**



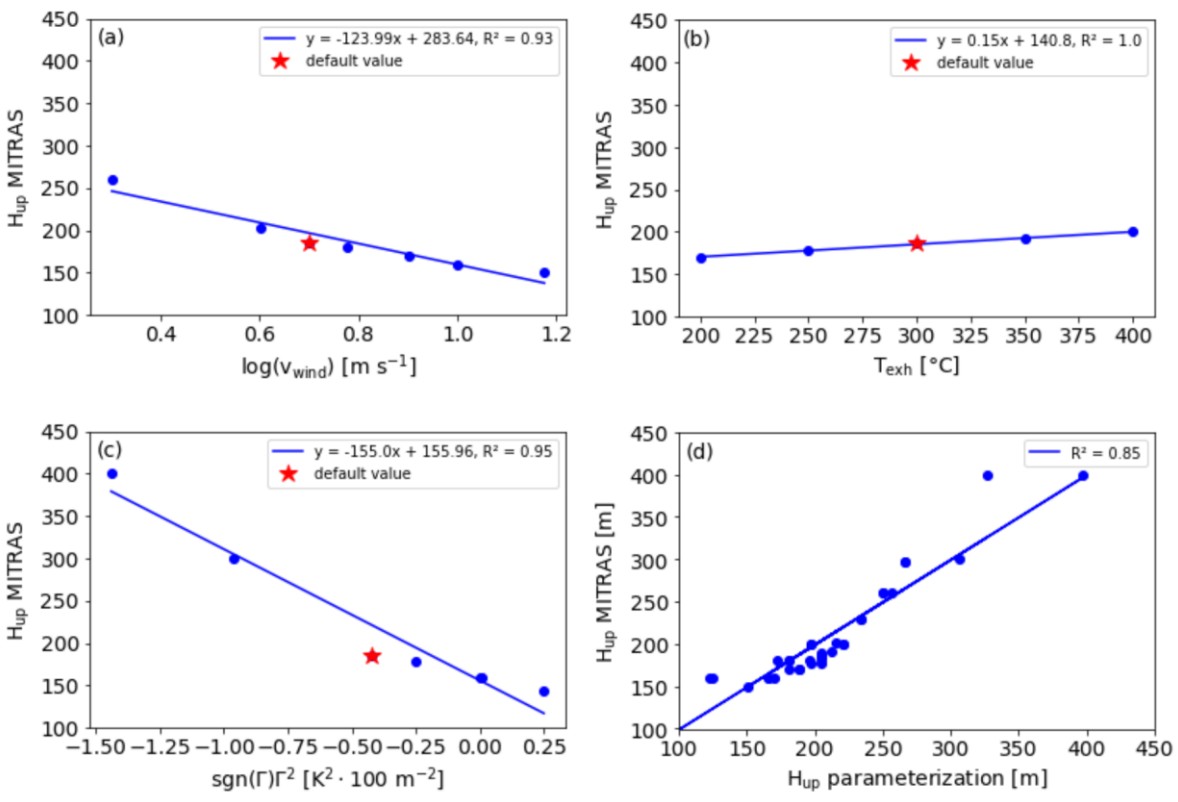

**Figure 5: Regression analysis for upper plume boundary heights calculated with MITRAS against (a) wind speed ($v_{wind}$), (b) exhaust temperature ($T_{exh}$) and (c) atmospheric stability ($\Gamma$). Panel (d) shows the regression of the upper plume boundary from MITRAS results against the parameterization.**


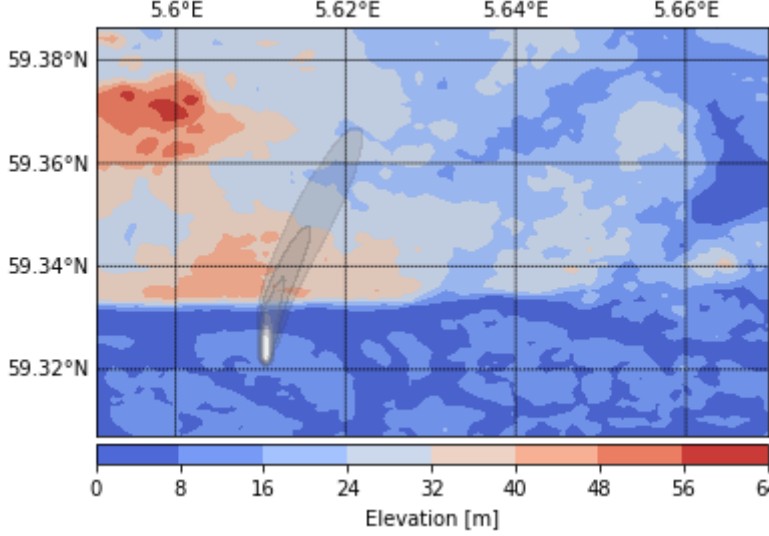




**Figure 6: Elevation map of the inner city of Hamburg in the EPISODE-CityChem model domain of 8 km · 8 km. An example ship plume is shown in grayscale.**

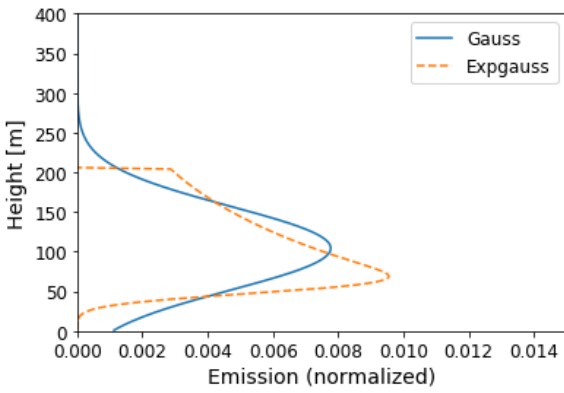

**Figure 7: Initial vertical concentration profiles for Gaussian, exponential Gaussian and single cell emissions under default conditions (see Table 1).**

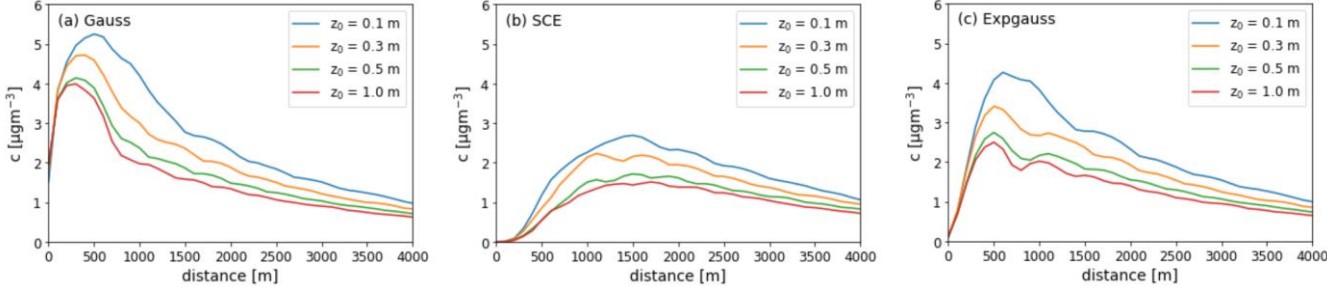

**Figure 8: Ground-level concentration profiles depending on the distance to the source for different roughness lengths and initial**
**emission distribution.**

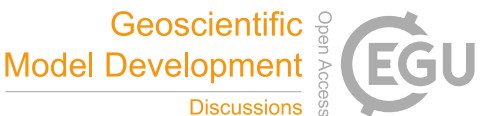

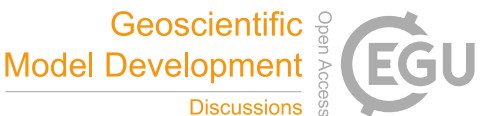

**Figure 9: (a−c) Ground-level concentration profiles depending on the distance to the source for different stabilities and initial emission distribution; (d) direct comparison of the effect of different emission distribution at Γ = -0.65 K · 100 m⁻¹.**



**Figure 10: (a−c)** Ground-level concentration profiles depending on the distance to the source for different wind speeds and initial emission distribution; **(d)** direct comparison of the effect of different emission distribution at $v_{wind} = 5$ m s$^{-1}$.



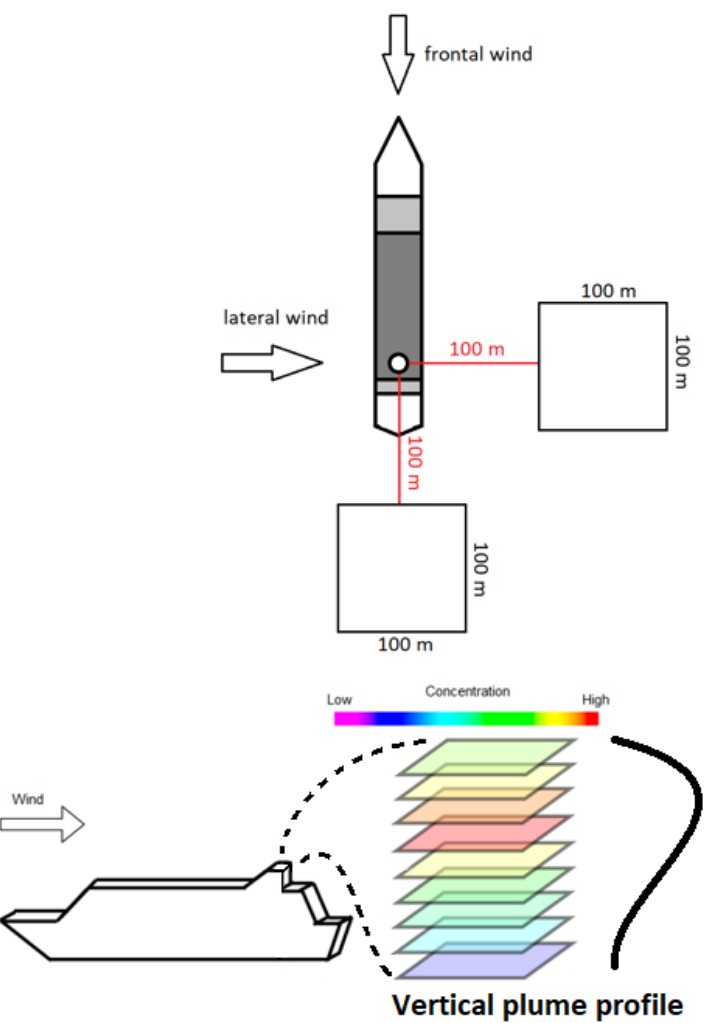

**Figure A1: Scheme for deriving the vertical plume concentration profile from MITRAS. Concentration values are derived from mean column values of 100 m · 100 m horizontal and 10 m vertical size in a distance of 100 m downwind from the ship to include plume rise and obstacle-induced turbulence in the emission profile. Adjusted from Badeke et al. (2021).**



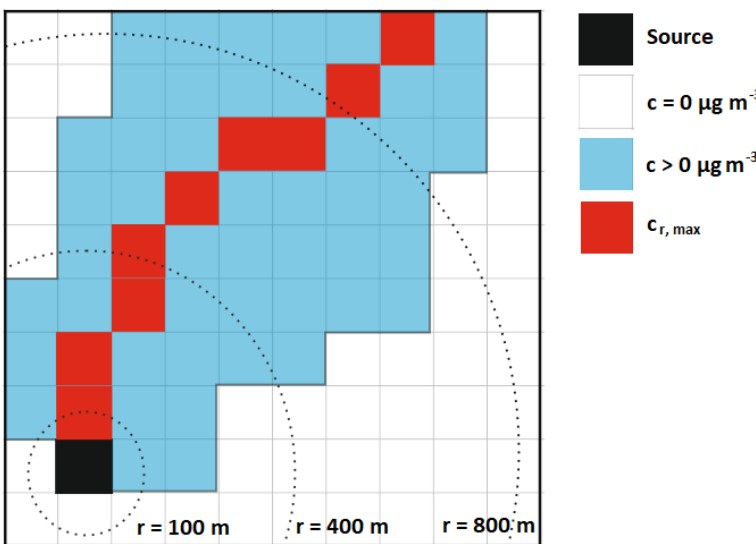

**Figure A2: Scheme for deriving ground-level concentration versus distance plots. Top view of the lowest model layer grid. The grid has a resolution of 100 m · 100 m. Blue cells are affected by the plume concentration, while white cells are not. For every radius of r = 100 m to r = 4000 m a circular function is applied to determine the highest concentration value along the perimeter. This is exemplarily shown for r = 100 m, 400 m and 800 m. Red cells show the resulting path of highest ground-level concentration.**



**Figure D1: Initial EPISODE-CityChem emission profiles under default input settings ($v_{wind}$ = 5 m s$^{-1}$, $v_{exit}$ = 10 m s$^{-1}$, $T_{exh}$ = 300 °C, $\Gamma$ = -0.65 K · 100 m$^{-1}$, $\phi$ = 0° and $z_{0, land}$ = 1 m) for all but one parameter. Panels (a) and (b) show effects of varying the stability while panels (c)–(f) show effects of varying the wind speed. Panel (d) represents full default conditions.**





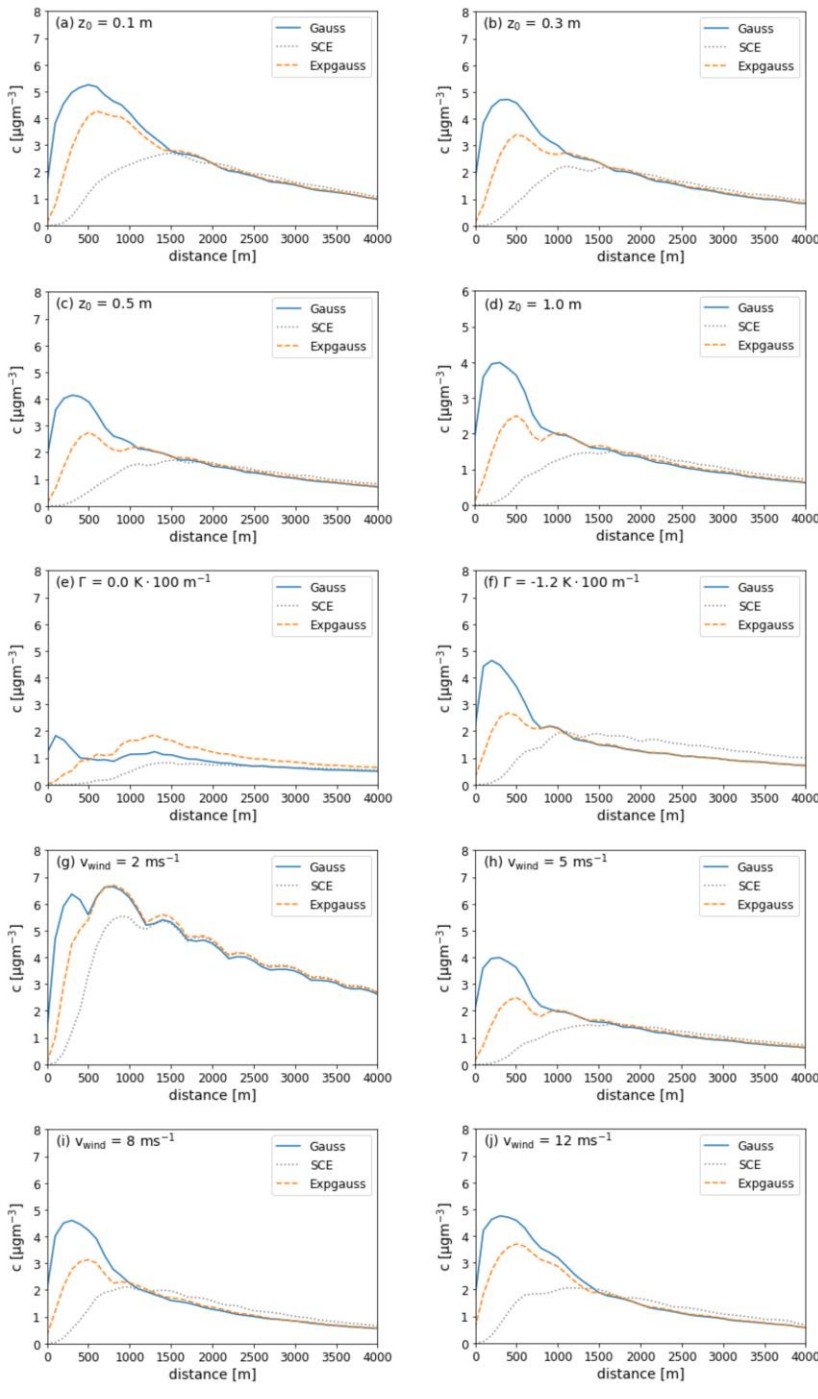

**Figure D2: Comparison for ground-level concentration values depending on the distance to the source for different settings and initial profiles. Default input settings ($v_{wind}$ = 5 m s$^{-1}$, $v_{exit}$ = 10 m s$^{-1}$, $T_{exh}$ = 300 °C, $\Gamma$ = -0.65 K · 100 m$^{-1}$, $\phi$ = 0° and $z_{0, land}$ = 1 m) were used for all but one parameter. Panels (a)−(d) vary roughness lengths over land, panels (e) and (f) vary atmospheric stability. Panels (g)−(j) vary wind speed. Panel (h) represents full default conditions.**





**Table 1: Input parameters for the MITRAS calculations. While varying one single input parameter in the investigated range, all others remain at default setting (adjusted from Badeke et al., 2021).**

| Input Parameter | Default Setting | Investigated Range |
|---|---|---|
| Ambient temperature at surface | 15 °C | None |
| Ambient temperature gradient | -0.65 K · 100 m$^{-1}$ | -1.2–0.5 K · 100 m$^{-1}$ |
| Wind speed at upper model boundary | 5 m s$^{-1}$ | 2–15 m s$^{-1}$ |
| Wind direction | 0° (frontal wind) | 0–90° |
| Surface roughness | ~ 0 m | None |
| Ship length | 246 m | None |
| Ship width | 30 m | None |
| Stack height | 52 m | None |
| Exit velocity | 10 m s$^{-1}$ | 4–12 m s$^{-1}$ |
| Exhaust temperature | 300 °C | 200–400 °C |

**Table 2: Overview of the EPISODE-CityChem setup**

| | |
|---|---|
| Horizontal domain size | 8 x 8 km² |
| Horizontal domain resolution | 100 m |
| Model grid coordinate system | WGS1984 Universal Transverse Mercator (UTM) Zone 32N |
| Vertical dimension | 30 layers |
| | Lowest 20 layers: 10 m |
| | Layers 21−30: step-wise increasing resolution up to 250 m |
| | Vertical top height: 1000 m |
| Meteorological inputs for MCWINDv1.2 | Ground temperature: 15 °C |
| | Wind direction: 180° |
| | Wind speed at stack height (50 m): 1−12 m s$^{-1}$ |
| | Atmospheric stability: -1.2−0.0 K · 100 m$^{-1}$ |
| | Cloud coverage: 100 % |
| Technical parameters of the ship (used for parameterization formulae) | Exhaust temperature: 300 °C |
| | Exit velocity: 10 m s$^{-1}$ |
| | Flow angle: 0° (frontal wind) |
| Surface roughness water | 0.001 m |





| Surface roughness land | 0.1 m−1.0 m |
|---|---|
| Emission rate | 1 g s⁻¹ |
| Emitted substance | $NO_x$ (95 % NO, 5 % $NO_2$) |
| | no reactions |
| Background chemistry | None |
| Emission type | Area emission |
| Vertical emission distribution | Gaussian profile |
| | One cell emission |
| | Exponentially modified Gaussian profile |


**Table B1: Data table for Gaussian regression analyses. Input data are wind speed at stack height of 50 m ($v_{wind}$), exit velocity ($v_{exit}$), exhaust temperature ($T_{exh}$) and wind direction ($\phi$), with 0° referring to frontal and 90° to lateral wind, and atmospheric stability ($\Gamma$). Results are mean ($\mu$), standard deviation ($\sigma$) and regression coefficient ($R^2$) for the regression analysis of MITRAS results against the fitted Gaussian functions and regression of fit against parameterization. The bold values in line no. 8 correspond to the default**
**settings.**

| Case # | $v_{wind}$ [m s⁻¹] | $v_{exit}$ [m s⁻¹] | $T_{exh}$ [°C] | $\phi$ [°] | $\Gamma$ [K · 100 m⁻¹] | $\mu_{fit}$ | $\sigma_{fit}$ | $R^2_{fit}$ | $\mu_{para}$ | $\sigma_{para}$ | $R^2_{fit \, vs. \, para}$ |
|---|---|---|---|---|---|---|---|---|---|---|---|
| 1 | 2.0 | 10 | 200 | 0 | -0.65 | 143 | 63.1 | 0.88 | 144 | 63.6 | 1.00 |
| 2 | 2.0 | 10 | 300 | 0 | -0.65 | 156 | 73.6 | 0.85 | 152 | 68.9 | 0.99 |
| 3 | 2.0 | 10 | 400 | 0 | -0.65 | 165 | 85.3 | 0.8 | 159 | 74.2 | 0.98 |
| 4 | 2.0 | 10 | 200 | 90 | -0.65 | 141 | 61.8 | 0.89 | 140 | 68.6 | 0.99 |
| 5 | 2.0 | 10 | 300 | 90 | -0.65 | 154 | 72.3 | 0.86 | 147 | 73.9 | 0.99 |
| 6 | 2.0 | 10 | 400 | 90 | -0.65 | 162 | 83.4 | 0.81 | 155 | 79.2 | 0.99 |
| 7 | 5.0 | 10 | 200 | 0 | -0.65 | 98 | 48.3 | 0.94 | 97 | 47.3 | 1.00 |
| **8** | **5.0** | **10** | **300** | **0** | **-0.65** | **106** | **53.7** | **0.90** | **104** | **52.6** | **1.00** |
| 9 | 5.0 | 10 | 400 | 0 | -0.65 | 112 | 56.7 | 0.88 | 112 | 57.9 | 1.00 |
| 10 | 5.0 | 10 | 200 | 90 | -0.65 | 89 | 53.9 | 0.96 | 92 | 52.3 | 1.00 |
| 11 | 5.0 | 10 | 300 | 90 | -0.65 | 95 | 59.4 | 0.94 | 100 | 57.6 | 1.00 |
| 12 | 5.0 | 10 | 400 | 90 | -0.65 | 101 | 62.4 | 0.93 | 107 | 62.9 | 0.99 |
| 13 | 8.0 | 4 | 200 | 0 | -0.65 | 68 | 38.9 | 0.97 | 69 | 36.5 | 1.00 |
| 14 | 8.0 | 4 | 300 | 0 | -0.65 | 78 | 41.8 | 0.97 | 76 | 41.8 | 1.00 |
| 15 | 8.0 | 4 | 400 | 0 | -0.65 | 84 | 43.7 | 0.97 | 84 | 47.1 | 1.00 |
| 16 | 8.0 | 4 | 200 | 90 | -0.65 | 65 | 40.2 | 0.99 | 64 | 41.5 | 1.00 |
| 17 | 8.0 | 4 | 300 | 90 | -0.65 | 70 | 43.8 | 0.99 | 72 | 46.8 | 1.00 |
| 18 | 8.0 | 4 | 400 | 90 | -0.65 | 75 | 47.4 | 0.98 | 79 | 52.1 | 0.99 |
| 19 | 5.0 | 10 | 250 | 0 | -0.65 | 102 | 50.5 | 0.91 | 101 | 50.0 | 1.00 |
| 20 | 5.0 | 10 | 350 | 0 | -0.65 | 110 | 54.8 | 0.89 | 108 | 55.3 | 1.00 |
| 21 | 4.0 | 10 | 300 | 0 | -0.65 | 115 | 55.9 | 0.86 | 116 | 56.6 | 1.00 |
| 22 | 6.0 | 10 | 300 | 0 | -0.65 | 96 | 49 | 0.93 | 95 | 49.4 | 1.00 |
| 23 | 8.0 | 10 | 300 | 0 | -0.65 | 83 | 43.1 | 0.97 | 80 | 44.2 | 1.00 |
| 24 | 10.0 | 10 | 300 | 0 | -0.65 | 73 | 42.4 | 0.97 | 68 | 40.3 | 0.99 |
| 25 | 5.0 | 4 | 300 | 0 | -0.65 | 100 | 50.1 | 0.91 | 101 | 50.2 | 1.00 |
| 26 | 5.0 | 8 | 300 | 0 | -0.65 | 104 | 52.3 | 0.90 | 103 | 51.8 | 1.00 |
| 27 | 5.0 | 12 | 300 | 0 | -0.65 | 108 | 54.1 | 0.89 | 106 | 53.4 | 1.00 |
| 28 | 5.0 | 10 | 300 | 0 | 0.50 | 97 | 35.9 | 0.93 | 104 | 37.4 | 0.97 |
| 29 | 5.0 | 10 | 300 | 0 | 0.10 | 100 | 40.4 | 0.93 | 104 | 42.7 | 0.99 |





| 30 | 5.0 | 10 | 300 | 0 | 0.00 | 102 | 41.8 | 0.92 | 104 | 44.0 | 1.00 |
| 31 | 5.0 | 10 | 300 | 0 | -0.50 | 105 | 50.2 | 0.91 | 104 | 50.6 | 1.00 |
| 32 | 5.0 | 10 | 300 | 0 | -0.98 | 96 | 50.1 | 0.84 | 104 | 57.0 | 0.97 |
| 33 | 5.0 | 10 | 300 | 0 | -1.20 | 93 | 50.0 | 0.85 | 104 | 59.9 | 0.95 |
| 34 | 10.0 | 4 | 200 | 90 | -0.98 | 49 | 45.2 | 0.98 | 52 | 41.9 | 1.00 |
| 35 | 15.0 | 10 | 300 | 0 | -0.65 | 58 | 37.7 | 0.96 | 47 | 33.0 | 0.93 |
| 36 | 15.0 | 4 | 200 | 90 | -1.20 | 40 | 40.2 | 0.99 | 31 | 37.5 | 0.97 |
| 37 | 5.0 | 10 | 300 | 45 | -0.65 | 102 | 53.1 | 0.88 | 103 | 54.1 | 1.00 |
| 38 | 5.0 | 10 | 300 | 60 | -0.65 | 98 | 55.8 | 0.90 | 102 | 55.1 | 1.00 |
| 39 | 5.0 | 10 | 300 | 30 | -0.65 | 103 | 51.8 | 0.88 | 104 | 53.3 | 1.00 |

**Table B2: Data table for Expgauss regression analyses. Input data are wind speed at stack height of 50 m ($v_{wind}$), exit velocity ($v_{exit}$), exhaust temperature ($T_{exh}$) and wind direction ($\phi$), with 0° referring to frontal and 90° to lateral wind, and atmospheric stability ($\Gamma$). Results are upper plume boundary height ($h_{up}$), shape parameters for the Expgauss function ($\lambda_1, \lambda_2, \lambda_3$) and regression coefficient (R²) for the regression analysis of MITRAS results against the fitted Expgauss functions and regression of fit against parameterization. The bold values in line no. 8 correspond to the default settings.**

| Case # | $v_{wind}$ [m s⁻¹] | $v_{exit}$ [m s⁻¹] | $T_{exh}$ [°C] | $\varphi$ [°] | $\Gamma$ [K · 100 m⁻¹] | $h_{up}$ [m] | $\lambda_{1, fit}$ | $\lambda_{2, fit}$ | $\lambda_{3, fit}$ | $R^2_{fit}$ | $h_{up, para}$ [m] | $\lambda_{1, para}$ | $\lambda_{2, para}$ | $\lambda_{3, para}$ | $R^2_{fit\ vs.\ para}$ |
|---|---|---|---|---|---|---|---|---|---|---|---|---|---|---|---|
| 1 | 2.0 | 10 | 200 | 0 | -0.65 | 229 | 0.0037 | 67.40 | 7.97 | 1.00 | 232 | 0.0033 | 66.69 | 13.32 | 0.99 |
| 2 | 2.0 | 10 | 300 | 0 | -0.65 | 261 | 0.0037 | 70.64 | 9.10 | 1.00 | 249 | 0.0033 | 68.99 | 11.97 | 0.99 |
| 3 | 2.0 | 10 | 400 | 0 | -0.65 | 298 | 0.0040 | 71.45 | 9.24 | 1.00 | 265 | 0.0033 | 71.29 | 10.62 | 0.99 |
| 4 | 2.0 | 10 | 200 | 90 | -0.65 | 229 | 0.0042 | 69.52 | 8.30 | 1.00 | 232 | 0.0033 | 63.83 | 21.60 | 0.94 |
| 5 | 2.0 | 10 | 300 | 90 | -0.65 | 261 | 0.0041 | 71.88 | 8.99 | 1.00 | 249 | 0.0033 | 66.13 | 20.25 | 0.95 |
| 6 | 2.0 | 10 | 400 | 90 | -0.65 | 298 | 0.0044 | 72.26 | 9.43 | 1.00 | 265 | 0.0033 | 68.43 | 18.90 | 0.96 |
| 7 | 5.0 | 10 | 200 | 0 | -0.65 | 170 | 0.0069 | 44.38 | 12.89 | 0.99 | 187 | 0.0093 | 45.72 | 13.32 | 0.99 |
| **8** | **5.0** | **10** | **300** | **0** | **-0.65** | **186** | **0.0065** | **47.26** | **8.53** | **0.99** | **203** | **0.0093** | **48.02** | **11.97** | **0.97** |
| 9 | 5.0 | 10 | 400 | 0 | -0.65 | 200 | 0.0067 | 50.44 | 7.71 | 0.99 | 220 | 0.0093 | 50.32 | 10.62 | 0.97 |
| 10 | 5.0 | 10 | 200 | 90 | -0.65 | 170 | 0.0100 | 38.17 | 22.62 | 0.99 | 187 | 0.0093 | 42.86 | 21.60 | 0.96 |
| 11 | 5.0 | 10 | 300 | 90 | -0.65 | 186 | 0.0081 | 37.76 | 19.32 | 1.00 | 203 | 0.0093 | 45.16 | 20.25 | 0.96 |
| 12 | 5.0 | 10 | 400 | 90 | -0.65 | 200 | 0.0101 | 43.48 | 20.53 | 0.96 | 220 | 0.0093 | 47.46 | 18.90 | 0.97 |
| 13 | 8.0 | 4 | 200 | 0 | -0.65 | 160 | 0.0203 | 35.07 | 22.18 | 0.95 | 164 | 0.0153 | 34.96 | 13.32 | 0.93 |
| 14 | 8.0 | 4 | 300 | 0 | -0.65 | 180 | 0.0222 | 44.83 | 27.57 | 0.95 | 180 | 0.0153 | 37.26 | 11.97 | 0.88 |
| 15 | 8.0 | 4 | 400 | 0 | -0.65 | 200 | 0.0196 | 48.01 | 26.54 | 0.96 | 197 | 0.0153 | 39.56 | 10.62 | 0.86 |
| 16 | 8.0 | 4 | 200 | 90 | -0.65 | 160 | 0.0340 | 41.13 | 32.31 | 0.98 | 164 | 0.0153 | 32.10 | 21.60 | 0.92 |
| 17 | 8.0 | 4 | 300 | 90 | -0.65 | 180 | 0.0249 | 39.42 | 31.09 | 0.98 | 180 | 0.0153 | 34.40 | 20.25 | 0.94 |
| 18 | 8.0 | 4 | 400 | 90 | -0.65 | 200 | 0.0143 | 33.43 | 25.32 | 0.99 | 197 | 0.0153 | 36.70 | 18.90 | 0.98 |
| 19 | 5.0 | 10 | 250 | 0 | -0.65 | 178 | 0.0067 | 45.52 | 9.49 | 0.99 | 195 | 0.0093 | 46.87 | 12.65 | 0.98 |
| 20 | 5.0 | 10 | 350 | 0 | -0.65 | 192 | 0.0065 | 48.94 | 7.93 | 0.99 | 212 | 0.0093 | 49.17 | 11.29 | 0.97 |
| 21 | 4.0 | 10 | 300 | 0 | -0.65 | 202 | 0.0061 | 52.65 | 5.38 | 1.00 | 215 | 0.0073 | 53.12 | 11.97 | 0.97 |
| 22 | 6.0 | 10 | 300 | 0 | -0.65 | 180 | 0.0084 | 44.24 | 13.51 | 1.00 | 194 | 0.0113 | 43.84 | 11.97 | 0.96 |
| 23 | 8.0 | 10 | 300 | 0 | -0.65 | 170 | 0.0139 | 41.17 | 21.32 | 0.97 | 180 | 0.0153 | 37.26 | 11.97 | 0.90 |
| 24 | 10.0 | 10 | 300 | 0 | -0.65 | 160 | 0.0147 | 33.76 | 19.98 | 0.96 | 169 | 0.0193 | 32.15 | 11.97 | 0.90 |
| 25 | 5.0 | 4 | 300 | 0 | -0.65 | 178 | 0.0069 | 44.62 | 9.69 | 1.00 | 203 | 0.0093 | 48.02 | 11.97 | 0.98 |
| 26 | 5.0 | 8 | 300 | 0 | -0.65 | 182 | 0.0067 | 46.44 | 8.87 | 0.99 | 203 | 0.0093 | 48.02 | 11.97 | 0.98 |
| 27 | 5.0 | 12 | 300 | 0 | -0.65 | 190 | 0.0063 | 48.17 | 8.17 | 0.99 | 203 | 0.0093 | 48.02 | 11.97 | 0.96 |
| 28 | 5.0 | 10 | 300 | 0 | 0.50 | 144 | 0.0038 | 51.29 | 6.51 | 1.00 | 76 | 0.0027 | 52.45 | 5.07 | 0.99 |
| 29 | 5.0 | 10 | 300 | 0 | 0.10 | 160 | 0.0056 | 50.69 | 7.19 | 0.99 | 122 | 0.0050 | 50.91 | 7.47 | 1.00 |
| 30 | 5.0 | 10 | 300 | 0 | 0.00 | 160 | 0.0046 | 49.92 | 6.63 | 1.00 | 124 | 0.0056 | 50.52 | 8.07 | 1.00 |
| 31 | 5.0 | 10 | 300 | 0 | -0.50 | 180 | 0.0058 | 47.76 | 7.87 | 1.00 | 171 | 0.0084 | 48.59 | 11.07 | 0.98 |
| 32 | 5.0 | 10 | 300 | 0 | -0.98 | 300 | 0.0110 | 45.59 | 9.28 | 0.99 | 305 | 0.0112 | 46.74 | 13.95 | 0.99 |
| 33 | 5.0 | 10 | 300 | 0 | -1.20 | 400 | 0.0112 | 44.00 | 10.61 | 1.00 | 396 | 0.0125 | 45.89 | 15.27 | 0.99 |
| 34 | 10.0 | 4 | 200 | 90 | -0.98 | 260 | 0.0227 | 19.24 | 26.65 | 1.00 | 254 | 0.0212 | 25.72 | 23.58 | 0.97 |
| 35 | 15.0 | 10 | 300 | 0 | -0.65 | 150 | 0.0180 | 25.48 | 16.36 | 0.99 | 149 | 0.0293 | 22.87 | 11.97 | 0.82 |
| 36 | 15.0 | 4 | 200 | 90 | -1.20 | 400 | 0.0343 | 17.80 | 30.34 | 1.00 | 325 | 0.0325 | 15.59 | 24.90 | 0.99 |
| 37 | 5.0 | 10 | 300 | 45 | -0.65 | 186 | 0.0076 | 45.65 | 7.90 | 1.00 | 203 | 0.0093 | 47.16 | 14.45 | 0.97 |
| 38 | 5.0 | 10 | 300 | 60 | -0.65 | 186 | 0.0085 | 43.08 | 13.41 | 0.99 | 203 | 0.0093 | 46.59 | 16.11 | 0.98 |
| 39 | 5.0 | 10 | 300 | 30 | -0.65 | 186 | 0.0067 | 45.22 | 5.25 | 1.00 | 203 | 0.0093 | 47.64 | 13.05 | 0.94 |





**Table C1: Quantitative representation of how strong input parameters affect the shape parameters for Gaussian and Expgauss fits.**
**Values in the table indicate the possible change that an input variable could cause on the concentration profile shape parameters**
**(while all other inputs remained at default conditions, comparable to effective ranges in Badeke et al., 2021). Bold values were used**
**for the parameterization as they had both a strong impact and a clear correlation.**

| Input variable | Range | Gaussian Fit | | Expgauss Fit | | | |
|---|---|---|---|---|---|---|---|
| | | $\mu$ | $\sigma$ | $\lambda_1$ | $\lambda_2$ | $\lambda_3$ | $h_{up}$ |
| Wind Speed | 2–15 m s$^{-1}$ | **98.0** | **35.9** | **0.0145** | **45.16** | 15.94 | **110 m** |
| Wind direction | 0–90° (frontal to lateral) | **11.0** | **7.6** | 0.0020 | 9.50 | **14.07** | 0 m |
| Exit velocity | 4–12 m s$^{-1}$ | **8.0** | **4.0** | 0.0006 | **3.55** | 1.52 | 12 m |
| Exhaust temperature | 200–400 °C | **14.0** | **8.4** | 0.0006 | **6.06** | **5.18** | **30 m** |
| Stability | -1.2–0.5 K · 100 m$^{-1}$ | 13.0 | **17.8** | **0.0074** | **7.29** | **4.10** | **257 m** |

**Table D1: Comparison of effects of different input variables and initial emission profiles on the ground-level concentration. Values**
**of $\Delta c_{max}$ correspond to the highest absolute differences. Their corresponding relative difference is added in parenthesis and the**
**distance of $\Delta c_{max}$ is given as well.**

| Variable | Variable range | $\Delta c_{max, Gauss}$ | $\Delta c_{max, SCE}$ | $\Delta c_{max, Expgauss}$ |
|---|---|---|---|---|
| Roughness length | 0.1 m−1.0 m | 0.1 m: 2.72 µg m$^{-3}$ (113 %) higher than 1.0 m at 700 m distance | 0.1 m: 1.26 µg m$^{-3}$ (88 %) higher than 1.0 m at 1400 m distance | 0.1 m: 2.29 µg m$^{-3}$ (128 %) higher than 1.0 m at 700 m distance |
| Stability | -1.2 K · 100 m$^{-1}$ −0.0 K · 100 m$^{-1}$ | -1.2 K · 100 m$^{-1}$: 3.16 µg m$^{-3}$ (241 %) higher than 0.0 K · 100 m$^{-1}$ at 200 m distance | -1.2 K · 100 m$^{-1}$: 1.45 µg m$^{-3}$ (302 %) higher than 0.0 K · 100 m$^{-1}$ at 900 m distance | -1.2 K · 100 m$^{-1}$: 2.0 µg m$^{-3}$ (378 %) higher than 0.0 K · 100 m$^{-1}$ at 200 m distance |
| Wind speed | 1 m s$^{-1}$−12 m s$^{-1}$ | 1 m s$^{-1}$: 9.12 µg m$^{-3}$ (374 %) higher than 3 m s$^{-1}$ at 200 m distance | 1 m s$^{-1}$: 9.63 µg m$^{-3}$ (1095 %) higher than 5 m s$^{-1}$ at 600 m distance | 1 m s$^{-1}$: 9.73 µg m$^{-3}$ (506 %) higher than 5 m s$^{-1}$ at 600 m distance |