# Peer review of "Effects of vertical ship exhaust plume distributions on urban pollutant concentration - a sensitivity study with MITRAS v2.0 and EPISODE-CityChem v1.4"

_Geoscientific Model Development, 2021_

## Author Comment (AC1)

Point-by-point response for the comments of reviewer #1.
The font color of the reviewer's comments is in black and our response is in blue.

**General comment**

This papers aims to investigate the effect of vertical distribution of ship plumes on modelling of ground level concentrations at small spatial scale, i.e. a few kilometres from the source. The topic is of interest even because the impact of ship emissions at berth in port cities is actually concentrated at small distance from the port. Considering that specific parameterizations to be included in models are provided, I believe that it is a topic suitable for the Journal. It must be said that in a previous paper of the same authors, a very similar input dataset was used and the same modelled ship, I believe that there are elements of novelty in this analysis, however authors are invited to better clarify this aspect. In addition, some aspects of discussion and interpretation is not very clear, see my specific comments, and a revision step will furnish benefits to this paper.

We appreciate the reviewer's opinion about the paper's content and suitability for the Journal. Thank you for your productive feedback.

**Specific comments**

Authors should explain better what are the elements of novelty compared to the previous paper because a similar modelling approach and the same input ranges are actually used.

The novelty of this study are the new parameterizations for vertical emission distribution based on MITRAS results. Badeke et al. (2021) analyzed the downward dispersion and pollution effects only in the near-field. This study investigates resulting differences in further distances (downwind inside the city). It is also an actual application of the microscale parameterization in a city-scale model. Using the same dataset allows a coherent use of both study results.

The following section has been added to the introduction (**lines 120-128**):

"In Eulerian city-scale models, the emission of a source like a stack are not necessarily inserted into only one grid cell, but can be vertically distributed to account for effects of plume rise and downward dispersion in the near-field. These initial emission profiles are herein defined as the relative vertical distribution of an emission value into one or multiple vertical grid cells. A Gaussian distribution, similar to the simple Gaussian plume models, would be the first guess for such a distribution. However, the results of Badeke et al. (2021), Bieser et al. (2011) and Brunner et al. (2019) led to the assumption, that for short ship stacks that are close to the obstacle itself, the downward dispersion may lead to a significantly different shape than a Gaussian distribution."

Lines 28-32. Here it would be useful to put a priority in nitrogen oxides that could be significantly impacted by shipping leading to overcomes of the legislation threshold in specific areas of port cities. SOx is certainly a ship-related pollutant but less relevant for maintaining local air quality standards. However, it could lead to formation of secondary aerosol (sulphate) that could have non negligible impacts at local and regional scales.

We agree to the comment and changed the order of substances to match with their relevance. We furthermore added information on current limit values.

Lines 33-37 now read:

"From an air quality perspective, the most problematic combustion products from ship exhaust are oxides of nitrogen ($NO_x = NO + NO_2$) and particulate matter (PM), followed by oxides of sulfur ($SO_x$), carbon monoxide (CO) and volatile organic compounds (VOC). In particular, the limit values for $NO_2$ of the EU directive 2008/50 (annual mean of 40 µg m$^{-3}$ and 24-hour mean of 200 µg m$^{-3}$) and target values from of the World Health Organization (annual mean of 25 µg m$^{-3}$ and 24-hour mean of 10 µg m$^{-3}$) are often not reached (European Union, 2008; World Health Organization, 2021)."

In the introduction, chain citations of a lot of papers together such as in lines 50-54 is not a good practice because it does not give to the reader any real clue of why these papers have been cited.

We removed the chain citation and explain the cited papers in more detail now.

Lines 57-63 now read:

"Epidemiological and health-related economic studies have been investigating the health effects of ship emissions intensively over the last 15 years and describing their impact in harbor cities. This includes exposure studies (Ramacher et al., 2019), assessments of degradation in human health (Eyring et al., 2010), impacts of shipping emissions on mortality (Lin et al., 2018) and premature deaths (Andersson et al., 2009; Broome et al., 2016; Corbett et al., 2007; Liu et al., 2016), effects of organic shipping pollutants on health (Zhang et al., 2019), benefits from low-sulfur fuels (Sofiev et al., 2018; Winebrake et al., 2009) as well as health-related external costs from international ship traffic (Brandt et al., 2013)."

In the introduction is given a brief overview of the impact of ships to local air quality. I would like to suggest to consider the recent review Contini and Merico (Atmosphere 2021, 12, 92) in which an global overview of these impacts is provided.

Thank you for the suggestion. We now cite the review paper of Contini and Merico in lines 83-85:

"The review of Contini and Merico (2021) gives another comprehensive overview on the current knowledge of maritime emission impacts on the air quality, health and projections of regulation effects and mitigation strategies."

Lines 89-93. I believe that the over prediction effect is not only due to plume rise, rather, the uniform distribution of emissions on such a large grid could influence results.

This statement was based directly on the outcomes from Ramacher et al. (2020). The large grid effect on chemical results has been mentioned in the paragraph above. It might also play a role in the study of Ramacher et al. (2020). Therefore, lines 100-107 have been slightly adjusted to:

"Overestimations and inaccurate chemical transformation rates can occur if ship emissions are instantaneously diluted in a large grid, which reduces nonlinear reaction rates with the hydroxyl radical (OH) and leads to a longer lifetime of $NO_x$ (von Glasow et al., 2003; Vinken et al., 2011). This error can be reduced by using high-resolution numerical models.

In the Hamburg harbor study from Ramacher et al. (2020), a comparison with measurements revealed an overprediction of modelled $NO_2$ close to the port area. In their study all shipping emissions were released into the lowest vertical layer of the model (10 m) as area sources on a 1 km · 1 km grid without including effects of plume rise which might have led to the overprediction, along with the resolution effect mentioned before."

Lines 264-265. This is not clear. Probably you mean that you modelled gases as passive tracers because you analyse small spatial and temporal scales so that transformations are limited?

For a realistic air quality study, chemical transformation are also relevant on this spatial and temporal scale. The focus of this study was primarily to investigate the pollutant distribution and not the transformation. If chemical transformations are considered as well, several other sensitivities would have had to be considered (e.g. background chemistry, other local sources, radiation, etc.). This was beyond the scope of this technical study. Our results are however applicable to all gases that might be seen as passive tracers.

Lines 294-300 now read:

"The resulting parameterization for the vertical concentration profile is integrated in the city scale model system EPISODE-CityChem (Hamer et al., 2020; Karl et al., 2019b). This three-dimensional Eulerian grid model is used to simulate the emission, transport, dispersion, photochemical transformation and deposition of pollutants on a city-scale. In this study, the focus lies on investigating the dilution of ship plumes under varying initial emission profiles. Chemical reactions are deactivated in this study, to make it applicable to any passive tracer gas. Also, the highly nonlinear $NO_x$-$O_3$ chemistry would need an inclusion of background chemistry, diurnal differences for photochemistry and other sources to model $NO_x$ concentrations precisely. This was beyond the scope of this study. Therefore, gases are modelled as passive tracers."

I suggest to include a discussion relative to the applicability to real cases. All the paper is based on modelling of a single ship and the influence of considering (or not considering) the plume rise is discussed for this specific cases. However, in real cases, there would be a mixture of ships in which several parameters relevant to plume rise (like exhaust temperature and momentum flux) are not known. The total emission itself is generally rather uncertain as you also mentioned. So my question is if this approach to take into account vertical distribution would have a relevant practical implication in real cases or the uncertainties are large enough that this is a second order effect?

Investigating the effect and relevance of this new emission distribution approach on pollution in a realistic air quality study will be presented in a future study.

We assume this effect to have a practical applicability, since more and more information on large ships are collected in databases and missing information might be extrapolated from similar ships.

The following paragraph has been added to the discussion section (lines 514-520):

"The author's assume that besides the variety of uncertainties, the results of this study have a relevant practical implication in real cases. Most importantly due to including the wind speed as a variable into the calculation of vertical emission profiles which has the largest impact on the emission distribution and resulting concentrations. Since wind speed measurements are widely available, an inclusion of wind speeds into the distribution function is possible in any real case scenario. Further uncertainties like technical parameters can be extracted from engine datasheets for individual ships and, if not available, be extrapolated from similar ships or engines. An important tool to derive these information for individual ships is the recently developed ship emission modelling system MoSES (Schwarzkopf et al., 2021)."

Title of section 5.5. Better "Comparison of the effects of different input variables".

Thank you, this correction has been adopted.

Line 444. There is a non-necessary parenthesis.

Thank you, this has been corrected.

---

## Author Comment (AC2)

Point-by-point response for the comments of reviewer #2.
The font color of the reviewer's comments is in black and our response is in blue.

**General comment**

The paper is a prosecution of a previous work (Badeke et al. 2021) aimed to investigate the vertical distribution of ship emissions. In this paper the authors study the influence of the vertical distribution of emissions on the ground-level pollutant concentrations at a few kilometres distance from the mooring point. The topic is of interest for an accurate assessment of the impact of ship emissions at berth in port cities. The paper deals with the development of an atmospheric dispersion model chain, for this reason it fits with the arguments treated in GMD journal.

I suggest the publication of the paper after the authors will reply to the following comments.

The authors appreciate the reviewer's opinion about the paper's suitability for the Journal andthe relevance of the topic. We are grateful for the suggestion for publication and applied several changes based on the reviewer's suggestions.

**Specific comments**

Methodology 116-120. The authors would better explain which obstacles have considered using MITRASv2.0. I guess they considered only the shape of the ship. What happens if ships with different shape are considered? Could the eventual presence of buildings near to the dock influence the presented results?

In Badeke et al. (2021), the two extrema cases stack only and medium-sized cruise ship have been investigated and the strongest effect on the downward disturbance has been numbered to be 31 % downward dispersion under stack-only conditions against 55 % when considering lateral flow and a cruise ship. The shape of a cruise ship is considered to be similar to a fully loaded container ship, therefore, we assume this is the range of effect from the ship shape.

In a future study, a modification of the $\lambda_2$ parameter is planned to make the Expgauss parameterization applicable to different-sized ships. This will not change the general shape of the distribution but shift the height along the vertical axis to account for smaller ships.

The effects of buildings near the dock can affect the concentration similarly (e.g. due to channeling or building wake effects), but have not been considered in the MITRAS based parameterization. For a very specific harbor scenario, this might be better reproduced by including the buildings and cranes inside a harbor. However, this would lead to a reduced applicability of the parameterizations to more general cases. Increased turbulent diffusion is therefore only included via the surface roughness values in EPISODE-CityChem.

Line 145 "No chemical reactions occur in the simulations." This assumption can be considered valid also in case of very low wind speed?

No, in a complete air quality study, chemical reactions need to be considered independent on the wind speed. In this study, the main focus lay on the effects of different distributions on the ground concentration. The emitted substance is NOx (95 % NO, 5 % $NO_2$) treated as a passive tracer gas. Adding the highly nonlinear NOx chemistry into this study would make the interpretation more difficult and need the inclusion of background chemistry and other sources to correctly calculate concentrations. This is planned for a future study.

Line 185 "No clear correlation was found for μ against the atmospheric stability, but a negative dependency has been found for stability against σ". The absence of correlation of μ against the atmospheric stability is a logical consequence of the range of vertical height considered. If it contains all the plume also in case of high convective conditions then the result is logical. If the authors agree with this interpretation they could include it in the text. Otherwise give their interpretation.

We assume this to be caused by the selected default conditions ($v_{wind}$ = 5 m/s, $T_{exh}$ = 300°C, $v_{exit}$ = 10 m/s and frontal flow for which effects of stability against mean and standard deviation of the Gaussian distribution have been tested (see Fig. 2i). Under these default conditions, stability has a very weak effect on the standard deviation. It might have an effect under low wind speed and convective conditions but still wind speed and exhaust temperature have a much stronger effect on the mean height of the initial emission profile in the near-field. The selected default conditions represent average meteorological and technical conditions for ships inside Hamburg harbor (Badeke et al., 2021).

For a better clarification, lines 204-206 now read:

"No clear correlation was found for μ against the atmospheric stability (Fig. 2 panel i). This means, that under otherwise default conditions, the atmospheric stability does not show a significant influence on the mean plume height. A negative dependency has been found for stability against σ."

Line 195 "Especially in cases of strong winds and stable atmospheric conditions, the simple Gaussian distribution delivers good results." But cases of strong winds are of less interest for the impact on air quality. This reduces significantly the value of the simple Gaussian distribution. It would be evidenced.

A very important comment to which we agree. Lines 220-224 have been adjusted:

"Especially in cases of strong winds and stable atmospheric conditions, the simple Gaussian distribution delivers good results. However, in cases of strong plume rise at neutral or instable atmospheric conditions, fitting concentration profiles with a simple Gauss can result in a poorer fitting quality of $R^2$ = 0.8 (e.g. case # 6 in Appendix B1). This reduces the applicability for Gaussian plume profiles especially in case of air quality studies, when situations of high concentration accumulation (e.g., due to low wind speed or strong downward dispersion) have to be evaluated."

Line 315. It is not clear how initial vertical concentration profiles were converted into vertical emission profiles in EPISODE-CityChem. Could the author explain this point?

The following section has been added to the introduction (**lines 120-128**):

"In Eulerian city-scale models, the emission of a source like a stack are not necessarily inserted into only one grid cell, but can be vertically distributed to account for effects of plume rise and downward dispersion in the near-field. These initial emission profiles are herein defined as the relative vertical distribution of an emission value into one or multiple vertical grid cells. A Gaussian distribution, similar to the simple Gaussian plume models, would be the first guess for such a distribution. However, the results of Badeke et al. (2021), Bieser et al. (2011) and Brunner et al. (2019) led to the assumption, that for short ship stacks that are close to the obstacle itself, the downward dispersion may lead to a significantly different shape than a Gaussian distribution."

Furthermore, Appendix A1 has been adjusted to clarify how vertical emission profiles were derived.

[Figure]

**Figure A1: Scheme for deriving the vertical plume concentration profile from MITRAS and transformation into emission profiles in EPISODE-CityChem. Dimensionless concentration values are derived from mean column values of 100 m · 100 m horizontal and 10 m vertical size in a distance of 100 m downwind from the ship to include plume rise and obstacle-induced turbulence. Normalization of the concentration profile and redistribution into the coarser EPISODE-CityChem grid is done to derive the vertical emission profile in EPISODE-CityChem. Adjusted and expanded from Badeke et al. (2021).**

It would be useful to introduce the definition of upper plume boundary height.

The upper plume boundary definition has been added in lines 267-270:

"The height at which the plume temperature equals the ambient temperature is herein defined as upper plume boundary height $h_{up}$. It was calculated based on the MITRAS model results and parameterized similar to the concentration profile functions. It can cause sharp concentration gradients in cases of a stable surrounding atmosphere."

I did not find in the paper the exact definition of initial concentration profiles. It is necessary to introduce this definition at the first time the authors discuss about "initial concentration profile"

This definition is now included in lines 120-123.

"In Eulerian city-scale models, the emission of a source like a stack are not necessarily inserted into only one grid cell, but can be vertically distributed to account for effects of plume rise and downward dispersion in the near-field. These initial emission profiles are herein defined as the relative vertical distribution of an emission value into one or multiple vertical grid cells."

It should also be clearer now with the adjustments on Fig. A1

The profile for the single cell emission model is reported in the caption of Fig. 7 but is not present in the diagram.

The profile is not shown in the figure, because it would be just a straight line or a point with normalized emission of 1.0 at the mean height of the Gaussian profile.

We adjusted the caption of Fig. 7:

"Figure 7: Initial vertical concentration emission profiles for Gaussian , and exponential Gaussian and single cell emissions under default conditions (see Table 1). The single cell emission profile lies at the mean height of the Gaussian profile with a normalized emission of 1.0 (not shown)."

A final observation. The authors evaluated, among others, the impact of surface roughness on pollutant ground-level concentration. Since the interest is focused on the local scale (few kilometres from the source) do they consider important or necessary a precise description of the buildings and the streets instead of the use of a simple parameter like the surface roughness?

We think that, if available, the use of a more complex obstacle-resolving model would lead to a better representation of turbulent effects and street canyons than a simple value like surface roughness. This is useful, if one is interested in a specific pollution situation of a certain city. However, these information are not widely available and processing time for this kind of analysis on a city-scale is very long. The surface roughness approach is more readily applicable to different cities. It can also be modified to include more land cover class information (and, therefore, roughness lengths) if available. One aim of this study was to make the parameterizations applicable in model scales that are not obstacle-resolving.